# Toward modular in-situ visualization in Earth System Models: the regional modeling system RegESM 1.1

Ufuk Utku Turuncoglu [1]

[1]Informatics Institute, Istanbul Technical University, 34469, Istanbul, Turkey

**Correspondence:** Ufuk Utku Turuncoglu (ufuk.turuncoglu@itu.edu.tr)

**Abstract.** The data volume produced by regional and global multi-component Earth System Models is rapidly increasing because of the improved spatial and temporal resolution of the model components, and the sophistication of the numerical models regarding represented physical processes and their complex non-linear interactions. In particular, very small time steps need to be defined in non-hydrostatic high-resolution modeling applications to represent the evolution of the fast-moving processes such as turbulence, extra-tropical cyclones, convective lines, jet streams, internal waves, vertical turbulent mixing and surface gravity waves. Consequently, the employed small time steps cause extra computation and disk input-output overhead in the modeling system even if today's most powerful high-performance computing and data storage systems are considered. Analysis of the high volume of data from multiple Earth System Model components at different temporal and spatial resolution also poses a challenging problem to efficiently perform integrated data analysis of the massive amounts of data when relying on the traditional post-processing methods today. This study mainly aims to explore the feasibility and added value of integrating existing in-situ visualization and data analysis methods within the model coupling framework. The objective is to increase interoperability between Earth System multi-component code and data processing systems by providing an easy-to-use, efficient, generic and standardized modeling environment. The new data analysis approach enables simultaneous analysis of the vast amount of data produced by multi-component regional Earth System Models during the runtime. The presented methodology also aims to create an integrated modeling environment for analyzing fast-moving processes and their evolution both in time and space to support a better understanding of the underplaying physical mechanisms. The state-of-art approach can also be employed to solve common problems in the model development cycle e.g. designing a new sub-grid scale parameterization that requires inspecting the integrated model behavior at a higher temporal and spatial scale simultaneously and supporting visual debugging of the multi-component modeling systems, which usually are not facilitated by existing model coupling libraries and modeling systems.

## 1 Introduction

The multi-scale and inherently coupled Earth System Models (ESMs) make them challenging to study and understand. Rapid developments in Earth system science, as well as in high-performance computing and data storage systems, have enabled fully coupled regional or global ESMs to better represent relevant processes, complex climate feedbacks, and interactions among the coupled components. In this context, regional ESMs are employed when the spatial and temporal resolution of the global

climate models are not sufficient to resolve local features such as complex topography, land-sea gradients and the influence of human activities in a smaller spatial scale. Along with the development of the modeling systems, specialized software libraries for the model coupling become more and more critical to reduce the complexity of the coupled model development and increase the interoperability, reusability, and efficiency of the existing modeling systems. Currently, the existing model
coupling software libraries have two main categories: couplers and coupling frameworks.

Couplers are mainly specialized in performing specific operations more efficiently and quickly such as coordination of components and interpolation among model components. For example, OASIS3 (Valcke, 2013) uses multiple executable approaches for coupling model components but sequentially performing internal algorithms such as sparse matrix multiplication (SMM) operation for interpolation among model grids become a bottleneck along with increased spatial resolution of the
model components. To overcome the problem, OASIS4 uses parallelism in its internal algorithms (Redler et al., 2010), and OASIS3-MCT (Craig et al., 2017) interfaced with the Model Coupling Toolkit (MCT; Jacob et al., 2005; Larson et al., 2005) provides a parallel implementation of interpolation and data exchange. Besides generic couplers like OASIS, domain-specific couplers such as Oceanographic Multi-purpose Software Environment (OMUSE; Pelupessy et al., 2017) that aims to provide a homogeneous environment for ocean modeling to make verification of simulation models with different codes and numerical
methods and Community Surface Dynamics Modeling System (CSDMS; Overeem et al., 2013) to develop integrated software modules for modeling of Earth surface processes are introduced.

A coupling framework is an environment for coupling model components through a standardized calling interface and aims to reduce the complexity of regular tasks such as performing spatial interpolation across different computational grids and transferring data among model components to increase the efficiency and interoperability of multi-component modeling
systems. Besides the synchronization of the execution of individual model components, a coupling framework can simplify the exchange of metadata related to model components and exchanged fields through the use of existing conventions such as CF (Climate and Forecast) convention. The Earth System Modeling Framework (ESMF) is one of the most famous examples of this approach (Theurich et al., 2016). The ESMF consists of a standardized superstructure for coupling components of Earth system applications through a robust infrastructure of high-performance utilities and data structures that ensure consistent
component behavior (Hill et al., 2004). The ESMF framework is also extended to include the National Unified Operational Prediction Capability (NUOPC) layer. The NUOPC layer simplifies component synchronization and run sequence by providing additional programming interface between coupled model and ESMF framework through the use of a NUOPC "cap". In this case, a NUOPC "cap" is a Fortran module that serves as the interface to a model when it is used in a NUOPC-based coupled system. The term "cap" is used because it is a small software layer that sits on top of a model code, making calls into it and
exposing model data structures in a standard way. In addition to generic modeling framework like ESMF, Modular System for Shelves and Coasts (MOSSCO; Lemmen et al., 2018) creates a state-of-art domain and process coupling system by taking advantage of both ESMF and Framework for Aquatic Biogeochemical Models (FABM; Bruggeman and Bolding, 2014) for marine coastal Earth system community.

The recent study of Alexander and Easterbrook (2015) to investigate the degree of modularity and design of the existing
global climate models reveals that the majority of the models use central couplers to support data exchange, spatial inter-

polation, and synchronization among model components. In this approach, direct interaction does not have to occur between individual model components or modules, since the specific coupler component manages the data transfer. This approach is also known as the hub-and-spoke method of building a multi-component coupled model. A key benefit of using a hub-and-spoke approach is that it creates a more flexible and efficient environment for designing sophisticated multi-component modeling system regarding represented physical processes and their interactions. The development of the more complex and high-resolution modeling systems leads to an increased demand for both computational and data storage resources. In general, the high volume of data produced by the numerical modeling systems may not allow storing all the critical and valuable information to use later, despite recent advances in storage systems. As a result, the simulation results are stored in a limited temporal resolution (i.e., monthly averages), which are processed after numerical simulations finished (post-processing). The poor representation of the results of numerical model simulations prevents to analyze the fast-moving processes such as extreme precipitation events, convection, turbulence and non-linear interactions among the model components in a high temporal and spatial scale with the traditional post-processing approach.

The analysis of leading high-performance computing systems reveals that the rate of disk input-output (I/O) performance is not growing at the same speed as the peak computational power of the systems (Ahern, 2012; Ahrens, 2015). The recent report of U.S. Department of Energy (DOE) also indicates that the expected rate of increase in I/O bandwidth (100 times) will be slower than the peak system performance (500 times) of the new generations of exascale computers (Ashby et al., 2010). Besides, the movement of large volumes of data across relatively slow network bandwidth servers fails to match the ultimate demands of data processing and to archive tasks of the present high-resolution multi-component ESMs. As a result, the traditional post-processing approach has become a bottleneck in monitoring and analysis of fast-moving processes that require very high spatial resolution, due to the present technological limitations in high-performance computing and storage systems (Ahrens et al., 2014). In the upcoming computing era, state-of-art new data analysis and visualization methods are needed to overcome the above limitations evocatively.

Besides the traditional data analysis approach, the so-called in-situ visualization and co-processing approaches allow researchers to analyze the output while running the numerical simulations simultaneously. The coupling of computation and data analysis helps to facilitate efficient and optimized data analysis and visualization pipelines and boosts the data analysis workflow. Recently, a number of in-situ visualization systems for analyzing numerical simulations of Earth system processes have been implemented. For instance, the ocean component of Model for Prediction Across Scales (MPAS) has been integrated with an image-based in-situ visualization tool to examine the critical elements of the simulations and reduce the data needed to preserve those elements by creating a flexible work environment for data analysis and visualization (Ahrens et al., 2014; O'Leary et al., 2016). Additionally, the same modeling system (MPAS-Ocean) has been used to study eddies in large-scale, high-resolution simulations. In this case, the in-situ visualization workflow is designed to perform eddy analysis at higher spatial and temporal resolutions than available with traditional post-processing facing storage size and I/O bandwidth constraints (Woodring et al., 2016). Moreover, a regional weather forecast model (Weather Research and Forecasting Model; WRF) has been integrated with in-situ visualization tool to track cyclones based on an adaptive algorithm (Malakar et al., 2012). Despite the lack of generic and standardized implementation for integrating model components with in-situ visualization tools, the

previous studies have shown that in-situ visualization can produce analyses of simulation results, revealing many details in an efficient and optimized way. It is evident that more generic implementations could facilitate smooth integration of the existing standalone and coupled ESMs with available in-situ visualization tools (Ahrens et al., 2005; Ayachit, 2015; Childs et al., 2012) and improve interoperability between such tools and non-standardized numerical simulation codes.

The main aim of this paper is to explore the added value of integrating in-situ analysis and visualization methods with a model coupling framework (ESMF) to provide in-situ visualization for easy to use, generic, standardized and robust scientific applications of Earth system modeling. The implementation allows existing ESMs coupled with the ESMF library to take advantage of in-situ visualization capabilities without extensive code restructuring and development. Moreover, the integrated model coupling environment allows sophisticated analysis and visualization pipelines by combining information coming from
multiple ESM components (i.e., atmosphere, ocean, wave, land-surface) in various spatial and temporal resolutions. Detailed studies of fundamental physical processes and interactions among model components are vital to the understanding of complex physical processes and could potentially open up new possibilities for the development of ESMs.

## 2   The design of the modeling system

The RegESM (Regional Earth System Model; 1.1) modeling system can use five different model components to support many
different modeling applications that might require detailed representation of the interactions among different Earth system processes (Fig. 1a-b). The implementation of the modeling system follows the hub-and-spoke architecture. The driver that is responsible for the orchestration of the overall modeling system resides in the middle and acts as a translator among model components (atmosphere, ocean, wave, river routing, and co-processing). In this case, each model component introduces its NUOPC cap to plug into the modeling system. The modeling system is validated in different model domains such as Caspian
Sea (Turuncoglu et al., 2013), Mediterranean Basin (Surenkok and Turuncoglu, 2015; Turuncoglu and Sannino, 2017), and Black Sea Basin.

### 2.1   Atmosphere models (ATM)

The flexible design of RegESM modeling system allows choosing a different atmospheric model component (ATM) in the configuration of the coupled model for a various type of application. Currently, two different atmospheric model is compatible
with RegESM modeling system: (1) RegCM4 (Giorgi et al., 2012), which is developed by the Abdus Salam International Centre for Theoretical Physics (ICTP) and (2) the Advanced Research Weather Research and Forecasting (WRF) Model (ARW; Skamarock et al., 2005), which is developed and sourced from National Center for Atmospheric Research (NCAR). In this study, RegCM 4.6 is selected as an atmospheric model component because the current implementation of WRF coupling interface is still experimental and does not support coupling with co-processing component yet, but the next version of the
modeling system (RegESM 1.2) will be able to couple WRF atmospheric model with co-processing component. The NUOPC cap of atmospheric model components defines state variables (i.e., sea surface temperature, surface wind components), rotates

the winds relative to Earth, apply unit conversions and perform vertical interpolation to interact with the newly introduced co-processing component.

### 2.1.1 RegCM

The dynamical core of the RegCM4 is based on the primitive equation, hydrostatic version of the National Centre for Atmospheric Research (NCAR) and Pennsylvania State University mesoscale model MM5 (Grell, 1995). The latest version of the model (RegCM 4.6) also supports non-hydrostatic dynamical core to support applications with high spatial resolutions (< 10 km). The model includes two different land surface models: (1) Biosphere-Atmosphere Transfer Scheme (BATS; Dickinson et. al., 1989) and (2) Community Land Model (CLM), version 4.5 (Tawfik and Steiner, 2011). The model also includes specific physical parameterizations to define air-sea interaction over the sea and lake (one-dimensional lake model; Hostetler et al., 1993). The Zeng Ocean Air-Sea Parameterization (Zeng et al., 1998) is extended to introduce the atmosphere model as a component of the coupled modeling system. In this way, the atmospheric model can exchange both two and three-dimensional fields with other model components such as an ocean, wave and river routing components that are active in an area inside of the atmospheric model domain as well as in-situ visualization component.

### 2.1.2 WRF

The WRF model consists of fully compressible non-hydrostatic equations, and the prognostic variables include the three-dimensional wind, perturbation quantities of pressure, potential temperature, geo-potential, surface pressure, turbulent kinetic energy and scalars (i.e., water vapor mixing ratio, cloud water). The model is suitable for a broad range of applications and has a variety of options to choose parameterization schemes for the planetary boundary layer (PBL), convection, explicit moisture, radiation, and soil processes to support analysis of different Earth system processes. The PBL scheme of the model has a significant impact on exchanging moisture, momentum, and energy between air and sea (and land) due to the used alternative surface layer options (i.e., drag coefficients) in the model configuration. A few modifications are done in WRF (version 3.8.1) model itself to couple it with RegESM modeling system. These modifications include rearranging of WRF time-related subroutines, which are inherited from the older version of ESMF Time Manager API (Application Programming Interface) that was available in 2009, to compile model with the newer version of ESMF library (version 7.1.0) together with the older version that requires mapping of time manager data types between old and new versions.

### 2.2 Ocean models (OCN)

The current version of the coupled modeling system supports two different ocean model components (OCN): (1) Regional Ocean Modeling System (ROMS revision 809; Shchepetkin and McWilliams, 2005; Haidvogel et al., 2008), which is developed and distributed by Rutgers University and (2) MIT General Circulation Model (MITgcm version c63s; Marshall et al., 1997a, b). In this case, ROMS and MITgcm models are selected due to their large user communities and different vertical grid representations. Although the selection of ocean model components depends on user experience and application, often

the choice of vertical grid system has a determining role in some specific applications. For example, the ROMS ocean model uses terrain following (namely s-coordinates) vertical grid system that allows a better representation of the coastal processes but MITgcm uses z levels generally used for applications that involve open oceans and seas. Similar to the atmospheric model component, both ocean models are slightly modified to allow data exchange with the other model components. In the current

version of the coupled modeling system, there is no interaction between wave and ocean model components, which could be crucial for some applications (i.e., surface ocean circulation and wave interaction) that need to consider the two-way interaction between waves and ocean currents. The exchange fields defined in the coupled modeling system between ocean and atmosphere strictly depend on the application and the studied problem. In some studies, the ocean model requires heat, freshwater and momentum fluxes to be provided by the atmospheric component, while in others, the ocean component retrieves

surface atmospheric conditions (i.e., surface temperature, humidity, surface pressure, wind components, precipitation) to calculate fluxes internally, by using bulk formulas (Turuncoglu et al., 2013). In the current design of the coupled modeling system, the driver allows selecting the desired exchange fields from the predefined list of the available fields. The exchange field list is a simple database with all fields that can be exported or imported by the component. In this way, the coupled modeling system can be adapted to different applications without any code customizations in both the driver and individual model components.

**2.2.1   ROMS**

The ROMS is a three-dimensional, free-surface, terrain-following numerical ocean model that solves the Reynolds-averaged Navier-Stokes equations using the hydrostatic and Boussinesq assumptions. The governing equations are in flux form, and the model uses Cartesian horizontal coordinates and sigma vertical coordinates with three different stretching functions. The model also supports second, third and fourth order horizontal and vertical advection schemes for momentum and tracers via its

preprocessor flags.

**2.2.2   MITgcm**

The MIT general circulation model (MITgcm) is a generic and widely used ocean model that solves the Boussinesq form of Navier-Stokes equations for an incompressible fluid. It supports both hydrostatic and non-hydrostatic applications with a spatial finite-volume discretization on a curvilinear computational grid. The model has an implicit free surface in the surface and

partial step topography formulation to define vertical depth layers. The MITgcm model supports different advection schemes for momentum and tracers such as centered second order, third-order upwind and second-order flux limiters to support a variety of applications. The model used in the coupled modeling system was slightly modified by ENEA to allow data exchange with other model components. The detailed information about the regional applications of the MITgcm ocean model is described in the study of Artale et al. (2010) using PROTHEUS modeling system specifically developed for the Mediterranean Sea.

## 2.3 Wave model (WAV)

Surface waves play a crucial role in the dynamics of PBL in the atmosphere and the currents in the ocean. Therefore, the wave component is included in the coupled modeling system to have a better representation of atmospheric PBL and surface conditions (i.e., surface roughness, friction velocity, wind speed). In this case, the wave component is based on WAM Cycle-4 (4.5.3-MPI). The WAM is a third-generation model without any assumption on the spectral shape (Monbaliu et al., 2000). It considers all the main processes that control the evolution of a wave field in deep water, namely the generation by wind, the nonlinear wave–wave interactions, and also white-capping. The model was initially developed by Helmholtz-Zentrum Geesthacht (GKSS, now HZG) in Germany. The original version of the WAM model was slightly modified to retrieve surface atmospheric conditions (i.e., wind speed components or friction velocity and wind direction) from the RegCM4 atmospheric model and to send back calculated surface roughness. In the current version of the modeling system, wave component cannot be coupled with the WRF model due to the missing modifications in the WRF side. In the RegCM4, the received surface roughness is used to calculate air-sea transfer coefficients and fluxes over sea using Zeng ocean air-sea parameterization (Zeng et al., 1998). In this design, it is also possible to define a threshold for maximum roughness length (the default value is 0.02 m) and friction velocity (the default value is 0.02 m) in the configuration file of RegCM4 to ensure the stability of the overall modeling system. Initial investigation of the added value of atmosphere-wave coupling in the Mediterranean Sea can be found in Surenkok and Turuncoglu (2015).

## 2.4 River routing model (RTM)

To simulate the lateral freshwater fluxes (river discharges) at the land surface and to provide river discharge to ocean model component, the RegESM modeling system uses Hydrological Discharge (HD, version 1.0.2) model developed by Max Planck Institute (Hagemann and Dumenil, 1998; Hagemann and Lydia, 2001). The model is designed to run in a fixed global regular grid with 0.5° horizontal resolution using daily time series of surface runoff and drainage as input fields. In that case, the model uses the pre-computed river channel network to simulate the horizontal transport of the runoff within model watersheds using different flow processes such as overland flow, baseflow and riverflow. The river routing model (RTM) plays an essential role in the freshwater budget of the ocean model by closing the water cycle between the atmosphere and ocean model components. The original version of the model was slightly modified to support interaction with the coupled model components. To close water cycle between land and ocean, model retrieves surface and sub-surface runoff from the atmospheric component (RegCM or WRF) and provides estimated river discharge to the selected ocean model component (ROMS or MITgcm). In the current design of the driver, rivers can be represented in two different ways: (1) individual point sources that are vertically distributed to model layers, and (2) imposed as freshwater surface boundary condition like precipitation (P) or evaporation minus precipitation (E-P). In this case, the driver configuration file is used to select the river representation type (1 or 2) for each river individually. The first option is preferred if river plumes need to be defined correctly by distributing river discharge vertically among the ocean model vertical layers. The second option is used to distribute river discharge to the ocean surface when there is a need to apply river discharge to a large areal extent close to the river mouth. In this case, a special algorithm implemented

in NUOPC cap of ocean model components (ROMS and MITgcm) is used to find affected ocean model grids based on the effective radius (in km) defined in the configuration file of the driver.

## 2.5 The driver: RegESM

The RegESM (version 1.1) is completely redesigned and improved version of the previously used and validated coupled atmosphere-ocean model (RegCM-ROMS) to study the regional climate of Caspian Sea and its catchment area (Turuncoglu et al., 2013). To simplify the design and to create more generic, extensible and flexible modeling system that aims to support easy integration of multiple model components and applications, the RegESM uses a driver to implement the hub-and-spoke approach. In this case, all the model components are combined using ESMF (version 7.1.0) framework to structure coupled modeling system. The ESMF framework is selected because of its unique online re-gridding capability, which allows the driver to perform different interpolation types (i.e., bilinear, conservative) over the exchange fields (i.e., sea surface temperature, heat and momentum fluxes) and the NUOPC layer. The NUOPC layer is a software layer built on top of the ESMF. It refines the capabilities of ESMF by providing a more precise definition of a component model and how components should interact and share data in a coupled system. The ESMF also provides the capability of transferring computational grids in the model component memory, which has critical importance in the integration of the modeling system with a co-processing environment (see also Sect. 3). The RegESM modeling system also uses ESMF and NUOPC layer to support various configuration of component interactions such as defining multiple coupling time steps among the model components. An example configuration of the four-component (ATM, OCN, RTM, and WAV) coupled modeling system can be seen in Fig. 2. In this case, the RTM component runs in a daily time step (slow) and interacts with ATM and OCN components, but ATM and OCN components can interact each other more frequently (fast) such as every three hours.

The interaction (also called as run sequences) among the model components and driver are facilitated by the connector components provided by NUOPC layer. Connector components are mainly used to create a link between individual model components and driver. In this case, the number of active components and their interaction determines the number of connector component created in the modeling system. The interaction between model components can be in two way: (1) bi-directional such as atmosphere and ocean coupled modeling system or (2) unidirectional such as atmosphere and co-processing modeling system. In the uni-directional case, the co-processing component does not interact with the atmosphere model and only process retrieved information; thus there is one connector component.

The RegESM modeling system can use two different types of time-integration coupling scheme between the atmosphere and ocean components: (1) explicit and (2) semi-implicit (or leap-frog) (Fig. 3). In the explicit type coupling, two connector components (ATM-OCN and OCN-ATM direction) are executed concurrently at every coupling time step and model components start and stop at the same model time (Fig. 3a). In the semi-implict coupling type (Fig. 3b), the ocean model receives surface boundary conditions from the atmospheric model at one coupling time step ahead of the current ocean model time. The semi-implicit coupling aimed at lowering the overall computational cost of a simulation by increasing stability for longer coupling time steps.

As described earlier, the execution of the model components is controlled by the driver. Both sequential and concurrent execution of the model components is allowed in the current version of the modeling system. If the model components and the driver are configured to run in sequence on the same set of PETs (Persistent Execution Threads), then the modeling system executes in a sequential mode. This mode is a much more efficient way to run the modeling system in case of limited computing resources. In the concurrent type of execution, the model components run in mutually exclusive sets of PETs, but the NUOPC connector component uses a union of available computational resources (or PETs) of interacted model components. By this way, the modeling system can support a variety of computing systems ranging from local servers to large computing systems that could include high-speed performance networks, accelerators (i.e., Graphics Processing Unit or GPU) and parallel I/O capabilities. The main drawback of concurrent execution approach is to assign correct amount of computing resource to individual model components, which is not an easy task and might require an extensive performance benchmark of a specific configuration of the model components, to achieve best available computational performance. In this case, a load-balancing analysis of individual components and driver play a critical role in the performance of the overall modeling system. For example, the LUCIA (Load-balancing Utility and Coupling Implementation Appraisal) tool can be used to collect all required information such as waiting time and calculation time of each system components for a load-balancing analysis in the OASIS3-MCT based coupled system.

In general, the design and development of the coupled modeling systems involve a set of technical difficulties that arise due to the usage of the different computational grids in the model components. One of the most common examples is the mismatch between the land-sea masks of the model components (i.e., atmosphere and ocean models). In this case, the unaligned land-sea masks might produce artificial or unrealistic surface heat and momentum fluxes around the coastlines, narrow bays, straits and seas. The simplest solution is to modify the land-sea masks of the individual model components manually to align them however, this requires time and is complex (especially when the horizontal grid resolution is high). Besides, the procedure needs to be repeated each time the model domain (i.e., shift or change in the model domain) or horizontal grid resolution is changed.

The RegESM modeling system uses customized interpolation technique that also includes extrapolation to overcome the mismatched land-sea mask problem for the interaction between atmosphere, ocean and wave components. This approach helps to create more generic and automatized solutions for the remapping of the exchange fields among the model components and enhance the flexibility of the modeling system to adapt to different regional modeling applications. There are three main stages in the customized interpolation technique: (1) finding destination grid points that the land-sea mask type does not match completely with the source grid (unmapped grid points; Fig. 4), (2) perform bilinear interpolation to transfer the exchange field from source to destination grid, and (3) perform extrapolation in destination grid to fill unmapped grid points that are found in first step.

To find the unmapped grid points, the algorithm first interpolates the field from source to destination grid (just over the sea) using a nearest-neighbor type interpolation (from Field_A to Field_B). Similarly, the same operation is repeated by using a bilinear type interpolation (from Field_A to Field_C). Then, the results of both interpolation (Field_B and Field_C) is compared to identify unmapped grid points for the bilinear interpolation (Fig. 4).

The field can then be interpolated from the source to the destination grid using a two-step interpolation approach. In the first step, the field is interpolated from source to destination grid using a bilinear interpolation. Then, nearest-neighbor type interpolation is used on the destination grid to fill unmapped grid points. One of the main drawbacks of this method is that the result field might include unrealistic values and sharp gradients in the areas of complex land-sea mask structure (i.e., channels, straits). The artifacts around the coastlines can be fixed by applying a light smoothing after interpolation or using more sophisticated extrapolation techniques such as the sea-over-land approach (Kara et al., 2007; Dominicis et. al., 2014), which are not included in the current version of the modeling system. Also, the usage of the mosaic grid along with second-order conservative interpolation method, which gives smoother results when the ratio between horizontal grid resolutions of the source and destination grids are high, can overcome unaligned land-sea mask problem. The next major release of ESMF library (8.0) will include the creep fill strategy (Kara et al., 2007) to fill unmapped grid points.

## 3 Integration of a co-processing component in RegESM modeling system

The newly designed modeling framework is a combination of the ParaView co-processing plugin – which is called Catalyst (Fabian et. al., 2011) – and ESMF library that is specially designed for coupling different ESMs to create more complex regional and global modeling systems. In conventional co-processing enabled simulation systems (single physical model component such as atmosphere along with co-processing support), the Catalyst is used to integrate ParaView visualization pipeline with the simulation code to support in-situ visualization through the use of application-specific custom adaptor code (Malakar et al., 2012; Ahrens et al., 2014; O'Leary et al., 2016; Woodring et al., 2016). A visualization pipeline is defined as a data flow network in which computation is described as a collection of executable modules that are connected in a directed graph representing how data moves between modules (Moreland, 2013). There are three types of modules in a visualization pipeline: sources (file readers and synthetic data generators), filters (transforms data), and sinks (file writers and rendering module that provide images to a user interface). The adaptor code acts as a wrapper layer and transforms information coming from simulation code to the co-processing component in a compatible format that is defined using ParaView/Catalyst and VTK (Visualization Toolkit) APIs. Moreover, the adaptor code is responsible for defining the underlying computational grid and associating them with the multi-dimensional fields. After defining computational grids and fields, ParaView processes the received data to perform co-processing to create desired products such as rendered visualizations, added value information (i.e., spatial and temporal averages, derived fields) as well as writing raw data to the disk storage (Fig. 5a).

The implemented novel approach aims to create a more generic and standardized co-processing environment designed explicitly for Earth system science (Fig. 5b). With this approach, existing ESMs, which are coupled with ESMF library using NUOPC interface, may benefit from the use of an integrated modeling framework to analyze the data flowing from the multi-component and multi-scale modeling system without extensive code development and restructuring. In this design, the adaptor code interacts with the driver through the use of NUOPC cap and provides an abstraction layer for the co-processing component. As discussed previously, the ESMF framework uses a standardized interface (initialization, run and finalize routines) to plug new model components into existing modeling system such as RegESM in an efficient and optimized way. To that end, the new

approach will benefit from the standardization of common tasks in the model components to integrate co-processing compo-nent with the existing modeling system. In this case, all information (grids, fields, and metadata) required by ParaView/Catalyst is received from the driver, and direct interaction between other model components and the co-processing component is not allowed (Fig. 5b). The implementation logic of the adaptor code is very similar to the conventional co-processing approach

(Fig. 5a). However, in this case, it uses the standardized interface of the ESMF framework and NUOPC layer to define the com-putational grid and associated two and three-dimensional fields of model components. The adaptor layer maps the field (i.e., *ESMF_Field*) and grid (i.e., *ESMF_Grid*) objects to their VTK equivalents through the use of VTK and co-processing APIs, which are provided by ParaView and co-processing plugin (Catalyst). Along with the usage of the new approach, the interop-erability between simulation code and in-situ visualization system are enhanced and standardized. The new design provides an

easy-to-develop, extensible and flexible modeling environment for Earth system science.

The development of the adaptor component plays an essential role in the overall design and performance of the integrated modeling environment. The adaptor code mainly includes a set of functions for the initialization (defining computational grids and associated input ports), run and finalize the co-processing environment. Similarly, the ESMF framework also uses the same approach to plug new model components into the modeling system as ESMF components. In ESMF framework, the

simulation code is separated into three essential components (initialization, run and finalize) and calling interfaces are triggered by the driver to control the simulation codes (i.e., atmosphere and ocean models). In this case, the initialization phase includes definition and initialization of the exchange variables, reading input (initial and boundary conditions) and configuration files and defining the underlying computational grid (step 1 in Fig. 6). The run phase includes a time stepping loop to run the model component in a defined period and continues until simulation ends (step 4 in Fig. 6). The time interval to exchange data

between model and co-processing component can be defined using coupling time step just like the interaction among other model components. According to the ESMF convention, the model and co-processing components are defined as a gridded component while the driver is a coupler component. In each coupling loop, the coupler component prepares exchange fields according to the interaction among components by applying re-gridding (except coupling with co-processing component), performing a unit conversion and common operations over the fields (i.e., rotation of wind field).

In the new version of the RegESM modeling system (1.1), the driver is extended to redistribute two and three-dimensional fields from physical model components to allow interaction with the co-processing component. In the initialization phase, the numerical grid of ESMF components is transformed into their VTK equivalents using adaptor code (step 3 in Fig. 6). In this case, *ESMF_Grid* object is used to create *vtkStructuredGrid* along with their modified parallel two-dimensional decomposition configuration, which is supported by ESMF/NUOPC grid transfer capability (Fig. 7). According to the design, each model

component transfers their numerical grid representation to co-processing component at the beginning of the simulation (step 1 in Fig. 6) while assigning independent two-dimensional decomposition ratio to the retrieved grid definitions. The example configuration in Figure 7 demonstrates mapping of 2x3 decomposition ratio (in x and y-direction) of ATM component to 2x2 in the co-processing component. Similarly, the ocean model transfers its numerical grid with 4x4 decomposition ratio to co-processing component with 2x2 (Fig. 7). In this case, ATM and OCN model components do not need to have the same geographical domain. The only limitation is that the domain of ATM model component must cover the entire OCN model

domain for an ATM-OCN coupled system to provide the surface boundary condition for OCN component. The main advantage of the generic implementation of the driver component is to assign different computational resources to the components. The computational resource with accelerator support (GPU) can be independently used by co-processing component to do rendering (i.e., iso-surface extraction, volume rendering, and texture mapping) and processing the high volume of data in an efficient

and optimized way. The initialization phase is also responsible for defining exchange fields that will be transferred among the model components and maps *ESMF_Field* representations as *vtkMultiPieceDataSet* objects in co-processing component (step 2-3 in Fig. 6). Due to the modified two-dimensional domain decomposition structure of the numerical grids of the simulation codes, the adaptor code also modifies the definition of ghost regions – a small subset of the global domain that is used to perform numerical operations around edges of the decomposition elements. In this case, the ghost regions (or

halo regions in ESMF convention) are updated by using specialized calls, and after that, the simulation data are passed (as *vtkMultiPieceDataSet*) to the co-processing component. During the simulation, the co-processing component of the modeling system also synchronizes with the simulation code and retrieves updated data (step 5 in Fig. 6) to process and analyze the results (step 6 in Fig. 6). The interaction between driver and the adaptor continues until the simulation ends (step 4, 5 and 6 in Fig. 6) and the driver continues to redistribute the exchange fields using *ESMF_FieldRedist* calls. The NUOPC cap of model

components also supports vertical interpolation of the three-dimensional exchange fields to height (from terrain-following coordinates of RegCM atmosphere model) or depth coordinate (from s-coordinates of ROMS ocean model) before passing information to the co-processing component. In this design, the vertical interpolation is introduced to have a consistency in the vertical scales and units of the data coming from the atmosphere and ocean components. Then, finalizing routines of the model and co-processing components are called to stop the model simulations and the data analysis pipeline that destroy the defined

data structure/s and free the memory (step 7-8 in Fig. 6).

## 4    Use case and performance benchmark

To test the capability of the newly designed integrated modeling system described briefly in the previous section, the three component (atmosphere, ocean, and co-processing) configuration of RegESM 1.1 modeling system is implemented to analyze category 5 Hurricane Katrina. Hurricane Katrina was the costliest natural disaster and has been named one of the five dead-

liest hurricanes in the history of the United States, and the storm is currently ranked as the third most intense United States land-falling tropical cyclone. After establishing itself in the southern Florida coast as a weak category 1 storm near 22:30 UTC 25 August 2005, it strengthened to a category 5 storm by 12:00 UTC 28 August 2005 as the storm entered the central Gulf of Mexico (GoM). The model simulations are performed over a 3-day period i.e. 27-30 Aug. 2005, which is the most intense period of the cyclone, to observe the evolution of the Hurricane Katrina and understand the importance of air-sea interaction

regarding its development and predictability. The next section mainly details the three-components configuration of the modeling system as well as the computing environment, preliminary benchmark results performed with limited computing resource (without GPU support), and analysis of the evolution of Hurricane Katrina.

## 4.1 Working environment

The model simulations and performance benchmarks are done on a cluster (SARIYER) provided by the National Center for High-Performance Computing (UHeM) in Istanbul, Turkey. The CentOS 7.2 operating system installed in compute nodes are configured with a two Intel Xeon CPU E5-2680 v4 (2.40GHz) processor (total 28 cores) and 128 GB RAM. In addition to the compute nodes, the cluster is connected to a high-performance parallel disk system (Lustre) with 349 TB storage capacity. The performance network, which is based on Infiniband FDR (56 Gbps) is designed to give the highest performance for the communication among the servers and the disk system. Due to the lack of GPU accelerators in the entire system, the in-situ visualization integrated performance benchmarks are done with the support of software rendering provided by Mesa library. Mesa is an open-source OpenGL implementation that supports a wide range of graphics hardware each with its back-end called a renderer. Mesa also provides several software-based renderers for use on systems without graphics hardware. In this case, ParaView is installed with Mesa support to render information without using hardware-based accelerators.

## 4.2 Domain and model configurations

The Regional Earth System Model (RegESM 1.1) is configured to couple atmosphere (ATM; RegCM) and ocean (OCN; ROMS) models with newly introduced the co-processing component (ParaView/Catalyst version 5.4.1) to analyze the evolution of Hurricane Katrina and to assess the overall performance of the modeling system. In this case, two atmospheric model domains were designed for RegCM simulations using offline nesting approach, as shown in Fig. 8. The outer atmospheric model domain (low-resolution, LR) with a resolution of 27-km is centered at 77.5°W, 25.0°N and covers almost entire the United States, the western part of Atlantic Ocean and north-eastern part of Pacific Ocean for better representation of the large-scale atmospheric circulation systems. The outer domain is enlarged as much as possible to minimize the effect of the lateral boundaries of the atmospheric model in the simulation results of the inner model domain. The horizontal grid spacing of inner domain (high-resolution, HR) is 3-km and covers the entire GoM and the western Atlantic Ocean to provide high-resolution atmospheric forcing for coupled atmosphere-ocean model simulations and perform cloud-resolving simulations. Unlike the outer domain, the model for the inner domain is configured to use the non-hydrostatic dynamical core (available in RegCM 4.6) to allow better representation local scale vertical acceleration and essential pressure features.

The lateral boundary condition for the outer domain is obtained from European Centre for Medium-Range Weather Forecasts (ECMWF) latest global atmospheric reanalysis (ERA-Interim project; Dee et. al., 2011), which is available at 6-h intervals at a resolution of 0.75°x0.75° in the horizontal and 37 pressure levels in the vertical. On the other hand, the lateral boundary condition of the inner domain is specified by the results of the outer model domain. Massachusetts Institute of Technology-Emanuel convective parameterization scheme (MIT-EMAN; Emanuel, 1991; Emanuel and Zivkovic-Rothman, 1999) for the cumulus representation along with sub-grid explicit moisture (SUBEX; Pal et al., 2000) scheme for large-scale precipitations are used for low-resolution outer domain.

As it can be seen in Fig. 8, the ROMS ocean model is configured to cover entire the GoM to allow better tracking of the Hurricane Katrina. In this case, the used ocean model configuration is very similar to the configuration used by Physical

Oceanography Numerical Group (PONG), Texas A&M University (TAMU), in which the original model configuration can be accessed from their THREDDS (Thematic Real-time Environmental Distributed Data Services) data server (TDS). THREDDS is a service that aims to provide access to an extensive collection of real-time and archived datasets, and TDS is a web server that provides metadata and data access for scientific datasets, using a variety of remote data access protocols. The ocean model has a spatial resolution of 1/36°, which corresponds to a non-uniform resolution of around 3 km (655 x 489 grid points) with highest grid resolution in the northern part of the domain. The model has 60 vertical sigma layers ($\theta_s = 10.0$, $\theta_b = 2.0$) to provide detailed representation of the main circulation patterns of the region and vertical tracer gradients. The bottom topography data of the GoM is constructed using the ETOPO1 dataset (Amante and Eakins, 2009), and minimum depth ($h_c$) is set to 400 m. The bathymetry is also modified so that the ratio of depth of any two adjacent columns does not exceed 0.25 to enhance the stability of the model and ensure hydrostatic consistency that prevents pressure gradient error. The Mellor-Yamada level 2.5 turbulent closure (MY; Mellor and Yamada, 1982) is used for vertical mixing, while rotated tensors of the harmonic formulation are used for horizontal mixing. The lateral boundary conditions for ROMS ocean model are provided by Naval Oceanographic Office Global Navy Coastal Ocean Model (NCOM) during 27-30 August 2005.

The model coupling time step between atmosphere and ocean model component is set to 1 hour but 6 minutes coupling time step is used to provide one-way interaction with co-processing component to study Hurricane Katrina in a very high temporal resolution. In the coupled model simulations, the ocean model provides SST data to the atmospheric model in the region where their numerical grids overlap. In the rest of the domain, the atmospheric model uses SST data provided by ERA-Interim dataset (prescribed SST). The results of the performance benchmark also include additional tests with smaller coupling time step such as 3 minutes for the interaction with the co-processing component. In this case, the model simulations for the analysis of Hurricane Katrina runs over three days, but only one day of simulation length is chosen in the performance benchmarks to reduce the compute time.

## 4.3 Performance benchmark

A set of simulations are performed with different model configurations to assess the overall performance of the coupled modeling system by focusing on the overhead of the newly introduced co-processing component (Table 1). The performance benchmarks include analysis of the extra overhead provided by the co-processing component, coupling interval between physical models and co-processing component under different rendering load such as various visualization pipelines (Table 1). In this case, same model domains that are described in the previous section (Sect. 4.2) are also used in the benchmark simulations. The LR atmospheric model domain includes around 900.000 grid points while the HR domain contains 25 million grid points to test scaling up to a large number of processors. In both cases, the ocean model configuration is the same, and it has around 19 million grid points. Besides the use of a non-hydrostatic dynamical core in the atmospheric model component in the HR case, the rest of the model configuration is preserved. To isolate the overhead of the driver from the overhead of the co-processing component, first individual model components (ATM and OCN) are run in standalone mode and then, the best-scaled model configurations regarding two-dimensional decomposition configuration are used in the coupled model simulations; CPL (two component case: atmosphere-ocean) and COP (three-component case: atmosphere, ocean and co-processing component). Due

to the current limitation in the integration of the co-processing component, the coupled model only supports sequential type execution when the co-processing component is activated, but this limitation will be removed in the future version of the modeling system (RegESM 2.0). As mentioned in the previous section, the length of the simulations is kept relatively short (1 day) in the benchmark analysis to perform many simulations with different model configurations (i.e., coupling interval, visualization pipelines and domain decomposition parameters).

In the benchmark results, the slightly modified version of the speed-up is used because the best possible sequential implementation of the utilized numerical model (standalone and coupled) does not exist for the used demonstration application and model configurations. In this case, the speed-up is defined as the ratio of the parallel execution time for the minimum number of processors required to run the simulation ($T_p(N_{min})$; based on 140 cores in this study) to the parallel execution time ($T_p(N)$; see Eq. 1).

$$S(N) = \frac{T_p(N_{min})}{T_p(N)} \tag{1}$$

The measured wall clock time and the calculated speed-up of standalone model components (ATM and OCN) can be seen in Fig. 9. In this case, two different atmospheric model configurations are considered to see the effect of the domain size and non-hydrostatic dynamical core in the benchmark results (LR and HR; Fig. 8). The results show that the model scales pretty well and it is clear that the HR case shows better scaling results than LR configuration of the atmospheric component (ATM) as expected. It is also shown that around 588 processors, which is the highest available compute resource, the communication among the processors dominate the benchmark results of LR case, but it is not evident in HR case that scales very well without any performance problem (Fig. 9a). Similar to the atmospheric model component, the ocean model (OCN) is also tested to find the best two-dimensional domain decomposition configuration (tiles in x and y-direction). As it can be seen from the Fig. 9b, the selection of the tile configuration affects the overall performance of the ocean model. In general, model scales better if the tile in the x-direction is bigger than the tile in the y-direction, but this is more evident in the small number of processors. The tile effect is mainly due to the memory management of Fortran programming language (column-major order) as well as the total number of active grid points (not masked as land) placed in each tile. The tile options must be selected carefully while considering the dimension of the model domain in each direction. In some tile configuration, it is not possible to run the model due to the underlying numerical solver and the required minimum ghost points. To summarize, the ocean model scales well until 588 cores with the best tile configurations indicated in Fig. 9b.

The performance of the two-component modeling system (CPL) can be investigated using the benchmark results of the standalone atmosphere and ocean models. Similar to the benchmark results of the standalone model components, the measured wall clock time and the calculated speed-up of the coupled model simulations are also shown in Fig. 10. In this case, the best two-dimensional decomposition parameters of the standalone ocean model simulations are used in the coupled model simulations (Fig. 9b). The overhead is calculated by comparing the CPL wall clock-time to the sum of the standalone OCN and ATM wall clock time as they run sequentially. The comparison of the standalone and coupled model simulations show that the driver component introduces additional 5-10% (average is 5% for LR and 6% for HR cases) overhead in the total

execution time, which slightly increases along with the used total number of processors. The extra overhead is mainly due to the interpolation (sparse matrix multiply performed by ESMF) and extrapolation along the coastlines to match land-sea masks of the atmosphere and ocean models and fill the unmapped grid points to exchange data (Fig. 4) and slightly increases along with increased number of cores as well as number of MPI communication between the model components (Fig. 9 and 10a).

To further investigate the overhead introduced by the newly designed co-processing component, the three-component modeling system (COP) is tested with three different visualization pipelines (P1, P2, and P3; Table 1) using two different atmospheric model configurations (LR and HR) and coupling interval (3 and 6 minutes with co-processing). In this case, the measured total execution time during the COP benchmark results also includes vertical interpolation (performed in ESMF cap of the model components) to map data from sigma coordinates to height (or depth) coordinates for both physical model components (ATM

and OCN).

As shown in Fig. 10b-d, the co-processing components require 10-40% extra execution time for both LR and HR cases depending on used visualization pipeline when it is compared with CPL simulations. The results also reveal that the fastest visualization pipeline is P3 and the slowest one is P1 for the HR case (Fig. 10b and d). In this case, the components are all run sequentially, and the performance of the co-processing component becomes a bottleneck for the rest of the modeling system

especially for the computing environment without GPU support like the system used in the benchmark simulations. It is evident that if the co-processing were run concurrently in a dedicated computing resource, the overall performance of the modeling system would be improved because of the simultaneous execution of the physical models and co-processing components. Table 1 also includes the execution time of the single visualization pipeline (measured by using *MPI_Wtime* call) isolated from the rest of the tasks. In this case, each rendering task gets 2-4 seconds for P1 and P2 cases and 7-15 seconds for the P3 case in

LR atmospheric model configuration. For HR case, P1 and P2 take around 17-80 seconds, and the P3 case is rendered in around 8-10 seconds. These results show that the time spent in the co-processing component (sending data to ParaView/Catalyst, and rendering to create the output) fluctuates too much and that this component does not present a predictable and stable behavior. It might be due to the particular configuration of the ParaView, which is configured to use software-based rendering to process data in CPUs and load in the used high-performance computing system (UHeM) even if the benchmark tests are repeated

multiple times.

In addition to the testing modeling system with various data processing load, a benchmark with increased coupling time step is also performed (see P23M in Fig. 10c). In this case, the coupling time step between physical model components and the co-processing component is decreased (from 6 minutes to 3 minutes) to produce output in doubled frame rate, but coupling interval between physical model components (ATM and OCN) are kept same (1 hour). The benchmark results show that

increased coupling time step also rises overhead due to the co-processing from 45% to 60% for HR case and pipeline P2 when it is compared with the results of two-component simulations (CPL; Fig. 10a). It is also shown that the execution time of co-processing enabled coupled simulations increase but the difference between P2 and P23M cases are reduced from 66% to 37% when the number of processors increased from 140 to 588.

In addition to the analysis of timing profiles of modeling system under different rendering load, the amount of data ex-

changed and used in the in-situ visualization case can be compared with the amount of data that would be required for offline

visualization at the same temporal frequency to reveal the added value of the newly introduced co-processing component. For this purpose, the amount of data exchanged with co-processing component is given in Table 1 for three different visualization pipelines (P1, P2, and P3). In co-processing mode, the data retrieved from model component memory (single time step) by the driver is passed to the ParaView/Catalyst for rendering. In addition to processing data concurrently with the simulation on co-processing component, the offline visualization (post-processing) consists of computations done after the model is run and requires to store numerical results in a disk environment. For example, 3-days long simulation with 6-minutes coupling interval produces around 160 GB data (720 time-step) just for a single variable from high-resolution atmosphere component (P1 visualization pipeline) in case of using offline visualization. With co-processing, the same analysis can be done by applying the same visualization pipeline (P1), which requires to process only 224 MB data stored in the memory, in each coupling interval. Moreover, storing results of three-day long, high-resolution simulation of RegCM atmosphere model (in netCDF format) for offline visualization requires around 1.5 TB data in case of using 6-minutes interval in the default configuration (7 x 3d fields and 28 x 2d fields). It is evident that the usage of co-processing component reduces the amount of data stored in the disk and allows more efficient data analysis pipeline.

Besides the minor fluctuations in the benchmark results, the modeling system with co-processing component scales pretty well to the higher number of processors (or cores) without any significant performance pitfalls in the current configuration. On the other hand, the usage of accelerator enabled ParaView configuration (i.e., using NVIDIA EGL library) and ParaView plugins with accelerator support such as NVIDIA IndeX volume rendering plugin and new VTK-m filters to process data on GPU will improve the benchmark result. The NVIDIA IndeX for ParaView Plugin enables large-scale and high-quality volume data visualization capabilities of the NVIDIA IndeX library inside the ParaView and might help to reduce time to process high-resolution spatial data (HR case). In addition to NVIDIA IndeX plugin, VTK-m is a toolkit of scientific visualization algorithms for emerging processor architectures such as GPUs (Moreland, 2016). The model configurations used in the simulations also write simulation results to the disk in netCDF format. In case of disabling of writing data to disk or configure the models to write data with large time intervals (i.e., monthly), the simulations with active co-processing component will run much faster and make the analysis of the model results in real time efficiently especially in live mode (see Section 5.1).

# 5 Demonstration application

The newly designed modeling system can analyze numerical simulation results in both in-situ (or live) and co-processing modes. In this case, a Python script, that defines the visualization pipeline, mainly controls the selection of the operating mode and is generated using ParaView, co-processing plugin. The user could also activate live visualization mode, just by changing a single line of code (need to set coprocessor.EnableLiveVisualization as True) in Python script. This section aims to give more detailed information about two different approaches by evaluating numerical simulation of Hurricane Katrina in both models to reveal the designed modeling system capability and its limitations.

## 5.1 Live visualization mode

While the live visualization designed to examine the simulation state at a specific point in time, the temporal filters such as ParticlePath, ParticleTracer, TemporalStatistics that are designed to process data using multiple time steps cannot be used in this mode. However, live visualization mode allows connecting to the running simulation anytime through the ParaView GUI in order to make detailed analysis by modifying existing visualization pipelines defined by a Python script. In this case, the numerical simulation can be paused while the visualization pipeline is modified and will continue to run with the revised one. It is evident that the live visualization capability gives full control to the user to make further investigation about the simulation results and facilitate better insight into the underlying physical process and its evolution in time.

The current version of the co-processing enabled modeling system can process data of multiple model components by using multi-channel input port feature of ParaView/Catalyst. In this case, each model has two input channels based on the rank of exchange fields. For example, atmospheric model component has *atm_input2d* and *atm_input3d* input channels to make processing available both two and three-dimensional exchange fields. The underlying adaptor code resides between the NUOPC cap of co-processing component and ParaView/Catalyst and provides two grid definitions (2d and 3d) for each model components for further analysis. In this design, the ParaView Co-processing Plugin is used to generate Python co-processing scripts, and user needs to map data sources to input channels by using predefined names such as *atm_input2d* and *ocn_input3d*. Then, adaptor provides the required data to co-processing component through each channel to perform rendering and data analysis in real time. The fields that are used in the co-processing component are defined by generic ASCII formatted driver configuration file (*exfield.tbl*), which is also used to define exchange fields among other model components such as atmosphere and ocean models. Fig. 11 shows a screenshot of live visualization of three-dimensional relative humidity field provided by the low-resolution atmospheric model component, underlying topography information, and vorticity of ocean surface that is provided by ocean model component.

## 5.2 Co-processing mode

In addition to live visualization mode that is described briefly in the previous section, ParaView/Catalyst also allows to render and store data using predefined co-processing pipeline (in Python) for further analysis. Co-processing mode can be used for three purposes: (1) the simulation output can be directed to the co-processing component to render data in batch mode and write image files to the disk, (2) added value information (i.e., vorticity from wind components, eddy kinetic energy from ocean current) can be calculated and stored in a disk for further analysis and (3) storing simulation output in a higher temporal resolution to process it later (post-processing) or create a representative dataset that can be used to design visualization pipeline for co-processing or live visualization modes. In this case, the newly designed modeling system can apply multiple visualization and data processing pipelines to the simulation results at each coupling time step to make a different set of analysis over the results of same numerical simulation for more efficient data analysis. The modeling system also facilitates multiple input ports to process data flowing from multiple ESM components. In this design, input ports are defined automatically by the co-processing component based on activated model components (ATM, OCN, etc.) and each model component has two

ports to handle two and three-dimensional grids (and fields) separately such as *atm_input2d*, *atm_input3d*, *ocn_input2d* and *ocn_input3d*.

To test the capability of the co-processing component, the evolution of Hurricane Katrina is investigated by using two different configurations of the coupled model (COP_LR and COP_HR) that are also used to analyze the overall computational

performance of the modeling system (see Section 4.3). In this case, both model configurations use the same configuration of OCN model component, but the different horizontal resolution of the ATM model is considered (27 km for LR and 3 km for HR cases).

Figure 12 shows 3-hourly snapshots of the model simulated clouds that are generated by processing three-dimensional relative humidity field calculated by the low-resolution version of the coupled model (COP_LR) using NVIDIA IndeX volume

rendering plugin as well as streamlines of Hurricane Katrina, which is calculated using three-dimensional wind field. The visualization pipeline also includes sea surface height and surface current from the ocean model component to make an integrated analysis of the model results. Figure 12a-b shows the streamlines that are produced by extracting the hurricane using ParaView *Threshold* filter. In this case, the extracted region is used as a seed to calculate backward and forward streamlines. In Figure 12c-e, sea surface height, sea surface current and surface wind vectors (10-meters) are shown together to give insight

about the interaction of ocean-related variables with the atmospheric wind. Lastly, the hurricane reaches to the land and starts to disappear due to increased surface roughness and lack of energy source (Fig. 12f). While the low-resolution of atmosphere model configuration is used, the information produced by the new modeling system enabled to investigate the evolution of the hurricane in a very high temporal resolution, which was impossible before. A day-long animation that is also used to create Figure 12 can be found as a supplemental video (Turuncoglu, 2018a).

In addition low-resolution model results revealing the evolution of the hurricane in a very high temporal resolution, low and high-resolution model results are also compared to see the added value of the increased horizontal resolution of the atmospheric model component regarding representation of the hurricane and its structure. To that end, a set of visualization pipelines are designed to investigate the vertical updraft in the hurricane, simulated track, precipitation pattern, and ocean state. In this case, two time snapshots are considered: (1) 28 August 2005 0000 UTC, at the early stage of the hurricane in Category 5 and (2)

29 August 2005 0000 UTC just before Katrina makes its third and final landfall near Louisiana–Mississippi border, where the surface wind is powerful, and surface currents had a strong onshore component (McTaggart-Cowan et al., 2007a, b). In the analysis of vertical structure, the hurricane is isolated based on the criteria of surface wind speed that exceeds $20\ ms^{-1}$ and the seed (basically set of points defined as *vtkPoints*) for ParaView *StreamTracerWithCustomSource* filter are defined dynamically using *ProgrammableFilter* as a circular plane with a radius of 1.2° and points distributed with 0.2° interval in both direction

(x and y) around the center of mass of the isolated region. Then, forward and backward streamlines of vorticity are computed separately to see inflow at low and mid levels and outflow at upper levels for both low (COP_LR; Fig. 13a, b, d and e) and high-resolution (COP_HR; Fig. 14a, b, d and e) cases. The analysis of simulations reveal that the vertical air movement shows higher spatial variability in high-resolution simulation (COP_HR) case even if the overall structure of the hurricane is similar in both cases. As expected, the strongest winds occur in a region forming a ring around the eyewall of the hurricane, which is

where the lowest surface pressure occurs.

Also, the analysis of cloud liquid water content shows that low and mid-levels of the hurricane have higher water content and spatial distribution of precipitation is better represented in high-resolution case (Fig. 14a-b and d-e), which is consistent with the previous modeling study of Trenberth et al. (2007).

It is also seen that the realistic principal and secondary precipitation bands around the eye of the hurricane are more apparent and well structured in the high-resolution simulation while the low-resolution case does not show those small-scale features (Fig. 13a-b and d-e). On the ocean side, the loop current, which is a warm ocean current that flows northward between Cuba and the Yucatan Peninsula and moves north into the Gulf of Mexico, loops east and south before exiting to the east through the Florida Straits and joining the Gulf Stream and is well defined by the ocean model component in both cases (Fig. 13c and f; Fig. 14c and f). The track of the hurricane is also compared with the HURDAT2 second-generation North Atlantic (NATL) hurricane database, which is the longest and most complete record of tropical cyclone (TC) activity in any of the world's oceans (Landsea and Franklin, 2013). In this case, the eye of the hurricane is extracted as a region with surface pressure anomaly is greater than 15 millibar (shown as a circular region near the best track). As it can be seen from the figures, Katrina moves over in the central Gulf, which is mainly associated with the loop current and persistent warm and cold eddies, and intensifies as it passes over the region due to the high ocean heat content in both simulation (Fig. 13c and f and Fig. 14c and f). The comparison of the low and high-resolution simulations also indicate that the diameter of hurricane-force winds at peak intensity is bigger in high-resolution simulation case at 29 August 2005 0000 UTC (Fig. 13f and Fig. 14f). An animation that shows the comparison of low and high-resolution model results can be found as a supplemental video (Turuncoglu, 2018b).

While the main aim of this paper is to give design details of the new in-situ visualization integrated modeling system and show its capability, the performance of the coupled modeling system to represent one of the most destructive hurricanes is very satisfactory especially for high-resolution case (COP_HR). Nonetheless, the individual components (atmosphere and ocean) of the modeling system can be tuned to have better agreement with the available observations and previous studies. Specifically for the analysis of the hurricane, a better storm tracking algorithm needs to be implemented using ParaView *Programmable Filter* by porting existing legacy Fortran codes for more accurate storm tracking in both live and co-processing mode.

## 6    Discussion of the concepts associated with interoperability, portability, and reproducibility

In the current design of the RegESM modeling system, the NUOPC cap of the co-processing component is designed to work with regional modeling applications that have specific horizontal grid (or mesh) types such as rectilinear and curvilinear grids. The newly introduced co-processing interface (NUOPC cap and adaptor code) now needs to be generalized to be compatible with other regional and global modeling systems coupled through ESMF and NUOPC layer. Specifically, the following issues need to be addressed to achieve better interoperability with existing modeling systems and model components: (1) redesigning the NUOPC cap of the co-processing component to support various global and regional mesh types such as Cubed-Sphere and unstructured Voronoi meshes, (2) extending the adaptor code to represent mesh and exchange fields provided by NUOPC cap using VTK and ParaView/Catalyst APIs, (3) adding support to co-processing interface for models with online nesting capability, and (4) adding support to have common horizontal grid definitions in the co-processing component and in the other

components to make integrated analysis of data (i.e., calculating air-sea temperature difference and correlation) produced by the various model components. Moreover, the co-processing interface can be tightly integrated with the NUOPC layer to provide a simplified API for designing new in-situ visualization integrated modeling systems in an efficient and standardized way. Besides the configuration used in this study, the RegESM modeling system is also tested with different model configurations such as coupling RegCM, MITgcm, and co-processing component to investigate air-sea interaction in the Black Sea basin. Initial results show that the co-processing component can also successfully process data flowing from different model configurations supported by RegESM.

When the diverse nature of high-performance computing systems, their hardware infrastructure (i.e., performance networks and storage systems) and software stacks (i.e., operating systems, compilers, libraries for inter-node communication and their different versions) are considered, realizing fully portable modeling system becoming increasingly crucial for the scientific community. In this case, the detailed examination of possible configurations of the modeling system and exiting computing environments can help to improve the flexibility and portability of the developed modeling system. Specifically for RegESM modeling system, the use case application and benchmark simulations revealed that the single executable approach (combining all model components into one program) used in the design of the modeling system can cause a portability problem when visualization and simulation run on concurrent resources. In the case of a homogeneous computing environment (all nodes with or without GPU support), the in-situ enabled modeling system runs without any particular problem because each MPI (Message Passing Interface) process has access to the same software and hardware resources. In contrast, some computing systems may have homogeneous underlying hardware and software stack (e.g., mixed servers with and without GPU support). As a result, the simulation with in-situ visualization would fail due to missing shared software libraries in underlying GPU. In this case, two approaches can be used to overcome the problem: (1) installation of required libraries on the entire system even on servers that do not have GPU support, and (2) restructuring modeling system to support two executables, one for the co-processing component and one for the physical model component. The second approach is considered a more generic and flexible solution and enhances the portability of the modeling system. It also allows implementing a loosely coupled in-situ visualization system and enables the use of specialized hardware (GPU and more memory) for rendering (Rivi et al., 2012). The main drawback of the loosely coupled in-situ visualization approach is that it requires transferring data over the network. As a result, the network performance can be a bottleneck for the modeling system, especially for high-resolution multi-component modeling applications.

When the complexity of regional and global ESMs are considered, developing fully reproducible, and portable modeling system is a challenging task and requires significant human interaction to keep track of detailed metadata and provenance information about the model, simulation and computing environment (in both software and hardware levels). The use of scientific workflows in Earth System science has demonstrated advantages in terms of metadata, provenance, error handling, and reproducibility in an automatized and standardized way (Turuncoglu et al., 2011, 2012; Turuncoglu, 2012). Additionally, the rapid development in the software container technology can help to design flexible and portable computing environments. Hence, the Docker container was implemented to examine the feasibility of using the container approach for our newly developed in-situ visualization integrated modeling system. A container is a standard unit of software that helps to create a software pack-

age including all its dependencies, which can then be ported from one computing environment to another without worrying about the underlying hardware infrastructure and software stack. It also enhances the numerical reproducibility of simulations by creating a standardized computing environment isolated from any dependencies. In this study, the Docker is selected as a container environment because it is widely adopted across the software industry and has a very active user community. Despite

the flexibility and easy to use nature of Docker containers, using specialized hardware such as NVIDIA GPUs, which require kernel modules and user-level libraries to operate, is not supported natively. Therefore, Docker container cannot access the underlying GPU resource to perform hardware level rendering for visualization and data analysis. To enable portable GPU-based containers, NVIDIA developed a special container that loads GPU driver into the container at lunch. As a part of this study, the newly developed RegESM modeling system was tested with both Docker (software rendering through the use of

Mesa library) and NVIDIA Docker (hardware based rendering). The initial results show that RegESM can take advantage of the container approach to create portable and reproducible modeling system in both in-situ and co-processing modes without considerable performance loss ( 5-10%). The added value of using NVIDIA Docker is that it enables to utilize the underlying GPU resource to perform rendering (i.e., representation of clouds using direct volume rendering method). More information about a Docker container for in-situ visualization enabled modeling system can be found in the dedicated GitHub repository

(see code availability section).

## 7 Summary and conclusions

In this study, the newly developed in-situ visualization integrated modeling system (RegESM 1.1) is used to demonstrate the feasibility and added value of the integrated modeling environment to analyze the high volume of data coming from a multi-component ESM in an integrated way, which was not possible before. In this case, ParaView/Catalyst is used as a co-processing

component to process and render data. The results of the selected use case (evolution of Hurricane Katrina) show that the co-processing component provides an easy-to-use and generic modeling and data analysis environment, which is independent of the underlying physical model components used. Moreover, it promotes the usage of co-processing capability with the existing ESMs coupled using ESMF framework and NUOPC layer, without significant code restructuring and development and helps to increase the interoperability between ESMs and ParaView, co-processing plugin (Catalyst). In the current implementation, the

prototype version of the adaptor code acts as an abstraction layer to simplify and standardize the regular tasks to integrate the simulation code with in-situ visualization and analysis environment. The driver is also responsible for redistributing the data to the co-processing component while preserving its numerical grid along with the support of vertical interpolation. Coupling of the co-processing component with the generic driver facilitates the definition of custom data processing pipelines (defined by Python scripts) and allows analysis of data originated from different components (i.e., atmosphere and ocean models) of

the RegESM modeling system in a very high temporal resolution. In this way, RegESM modeling system can be used to study various physical processes (i.e., extreme precipitation events, air-sea interaction, convection, and turbulence) that could not be analyzed with the traditional post-processing approaches.

While the results of the in-situ visualization integrated modeling system are encouraging, the co-processing component will be extended to support different regional and global computational grid representations supported by ESMF library such as unstructured meshes for having a generic adaptor for various model applications. Additionally, we are currently exploring: (1) the way to optimize the transfer of grid features and mapping of exchange fields to enhance the overall performance of the modeling environment in terms of memory usage and computational efficiency especially for very high-resolution applications (< 3 km), (2) possibility of automatic detection of accelerators (GPUs) through the use of driver component and assigning available GPU resources automatically to the co-processing component for rendering, (3) improving modeling system and co-processing component to allow nested applications (both atmosphere and ocean), (4) developing more application of the integrated modeling environment (possibly with other ocean and atmosphere components such as WRF and MITgcm) to analyze different physical processes such as air-sea interactions in upwelling regions under extreme atmospheric forcing conditions.

*Code availability.* The RegESM modeling system is open source and available under the MIT License, making it suitable for community usage. The license allows modification, distribution, private and commercial uses. The source code for all versions of RegESM driver including 1.1 is distributed through the public code repository hosted by GitHub (https://github.com/uturuncoglu/RegESM). The user guide and detailed information about the modeling system are also distributed along with the source code in the same code repository. The RegESM source code includes the required code patches for the individual model components to be used as a component in the modeling system. On the other hand, the source code of individual model components such as the ocean, wave, and river routing components and co-processing tool (ParaView/Catalyst) used in the modeling system are distributed mainly by their home institutes that might apply different licensing types. The reader who wants to get more information about the individual model components and their license type can refer to their websites. The release version 1.1 is permanently archived on Zenodo and accessible under the digital object identifier doi:10.5281/zenodo.1307212. The demo configuration of the modeling system that is used in NVIDIA GPU Technology Conference (GTC) 2018 is also permanently archived on Zenodo and accessible under the digital object identifier doi:10.5281/zenodo.1474753. The repository also includes detailed information about the installation of the individual components of the modeling system, third-party libraries, and commands to create Docker container.

*Competing interests.* The author declares that he has no conflict of interest.

*Acknowledgements.* This study has been supported by a research grant (116Y136) provided by The Scientific and Technological Research Council of Turkey (TUBITAK). The computing resources used in this work were provided by the National Center for High Performance Computing of Turkey (UHeM) under grant number 5003082013 and 5004782017. The Quadro K5200 used for the development of the prototype version of the modeling system was donated by the NVIDIA Corporation as part of the Hardware Donation Program. The author extends his grateful thank to Rocky Dunlap and Robert Oehmke from NOAA/ESRL, and CIRES, Boulder, Colorado, Gerhard Theurich from Science Applications International Corporation, McLean, Virginia and Andy C. Bauer from Kitware Inc., USA and Mahendra Roopa from NVIDIA for their very useful suggestions and comments.

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

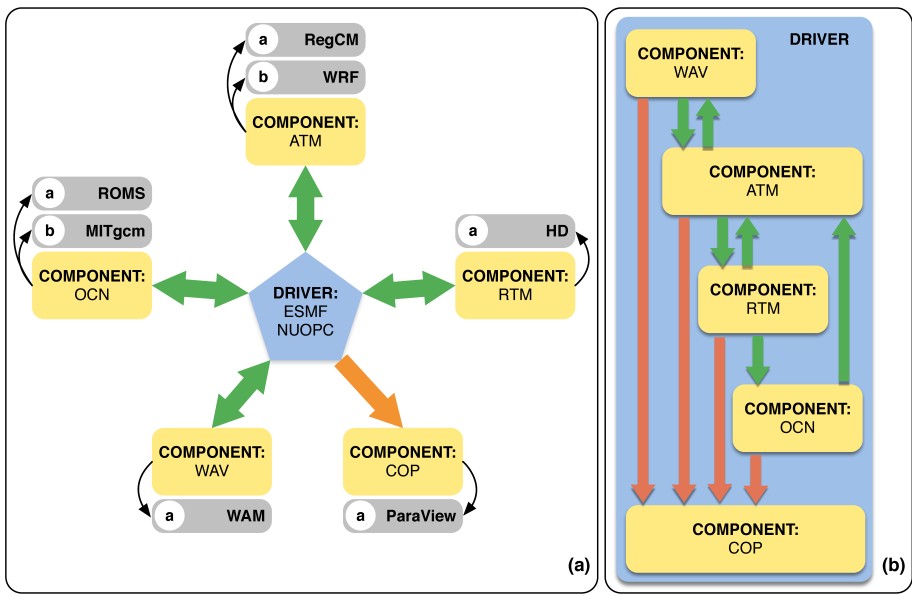

**Figure 1.** Design of the RegESM coupled modeling system: (a) model components including co-processing component, (b) their interactions (orange arrows represent the redistribution and green arrows shows interpolation).

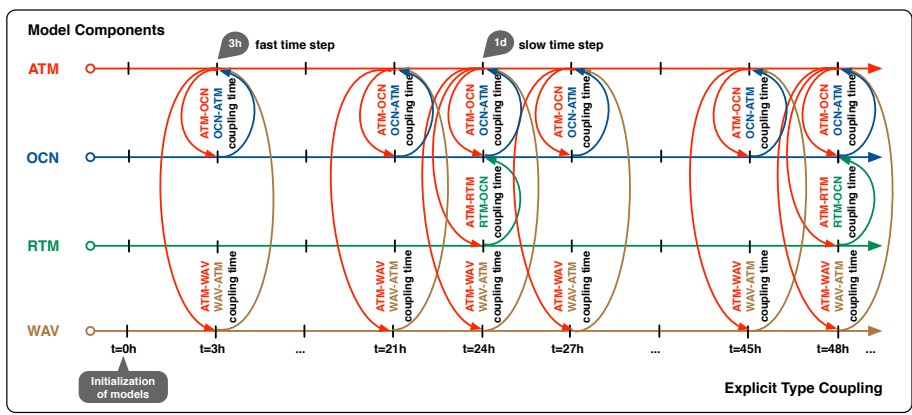

**Figure 2.** The run sequence of model components in case of explicit type coupling. In this case, the fast coupling time step is used for the interaction between the atmosphere, ocean and wave components. The slow coupling time step is only used to interact with the river routing component.

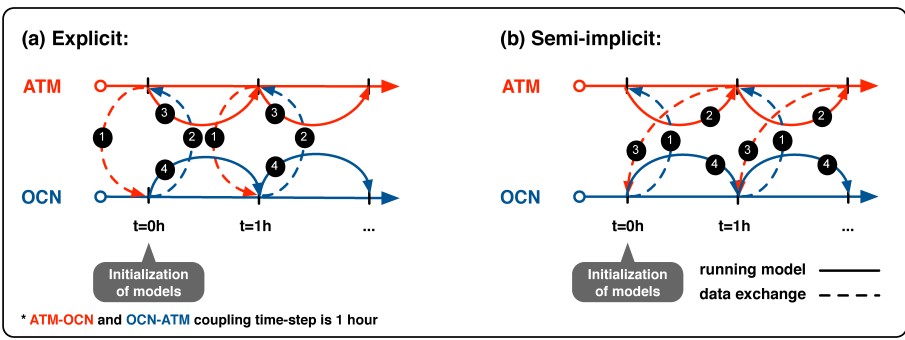

**Figure 3.** Schematic representation of (a) explicit and (b) semi-implicit model coupling between two model components (atmosphere and ocean). The numbers indicate the execution orders, which is initialized in each coupling interval.

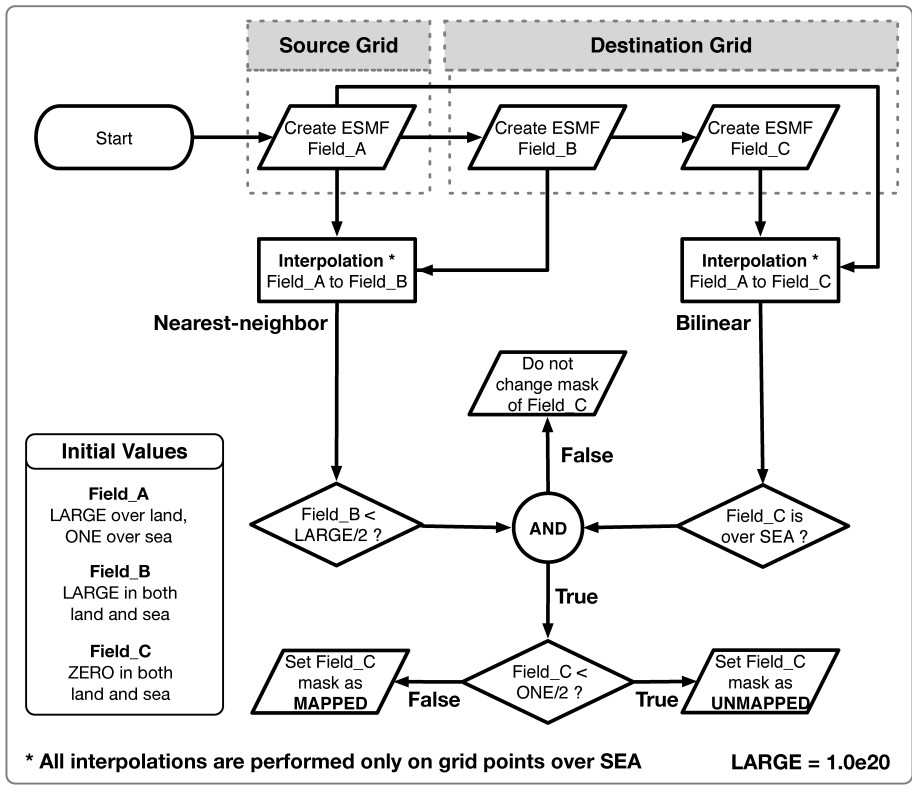

**Figure 4.** Processing flow chart of the algorithm to find mapped and unmapped grid points for two-step interpolation.

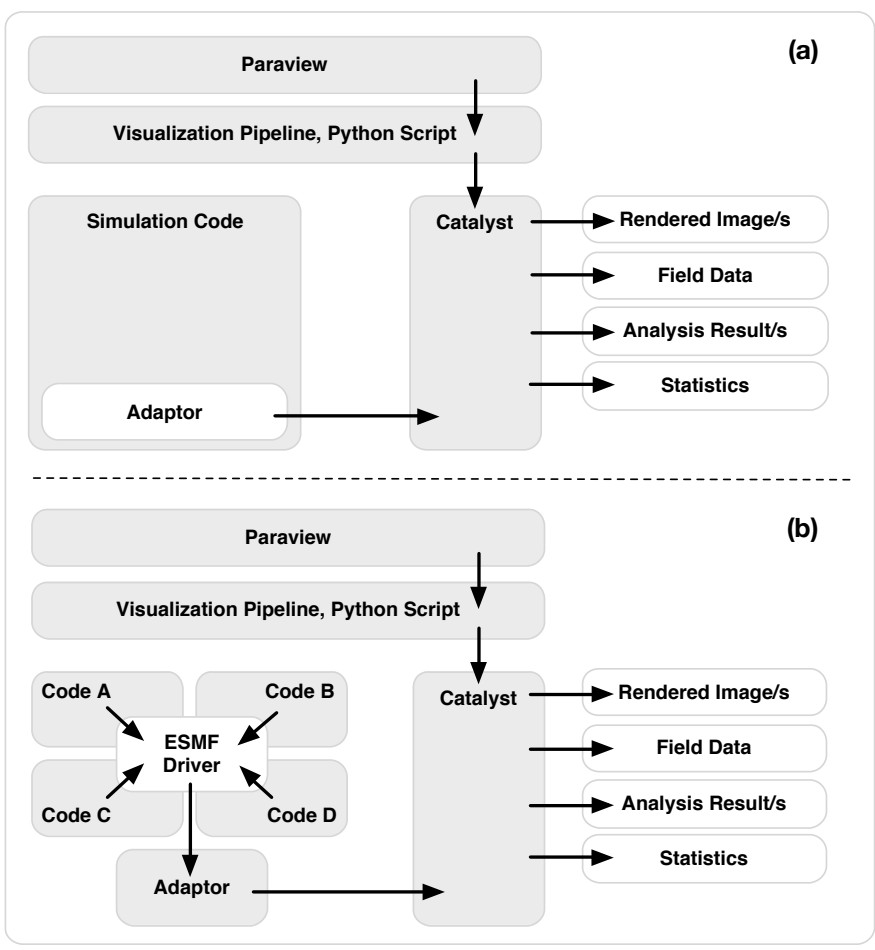

**Figure 5.** Comparison of the (a) conventional and (b) ESMF integrated in-situ visualization system.

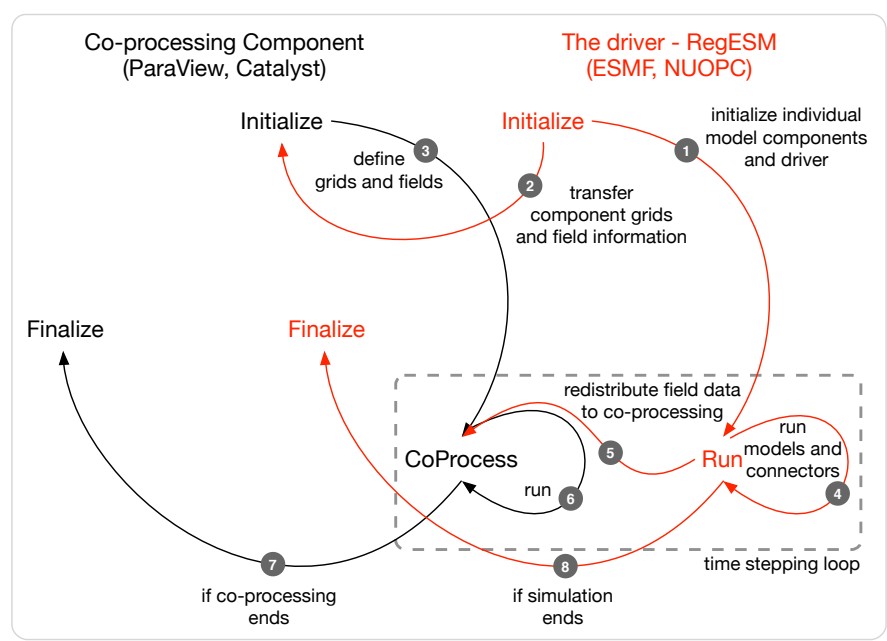

**Figure 6.** The interaction between driver defined by ESMF, NUOPC and co-processing component (Paraview/Catalyst).

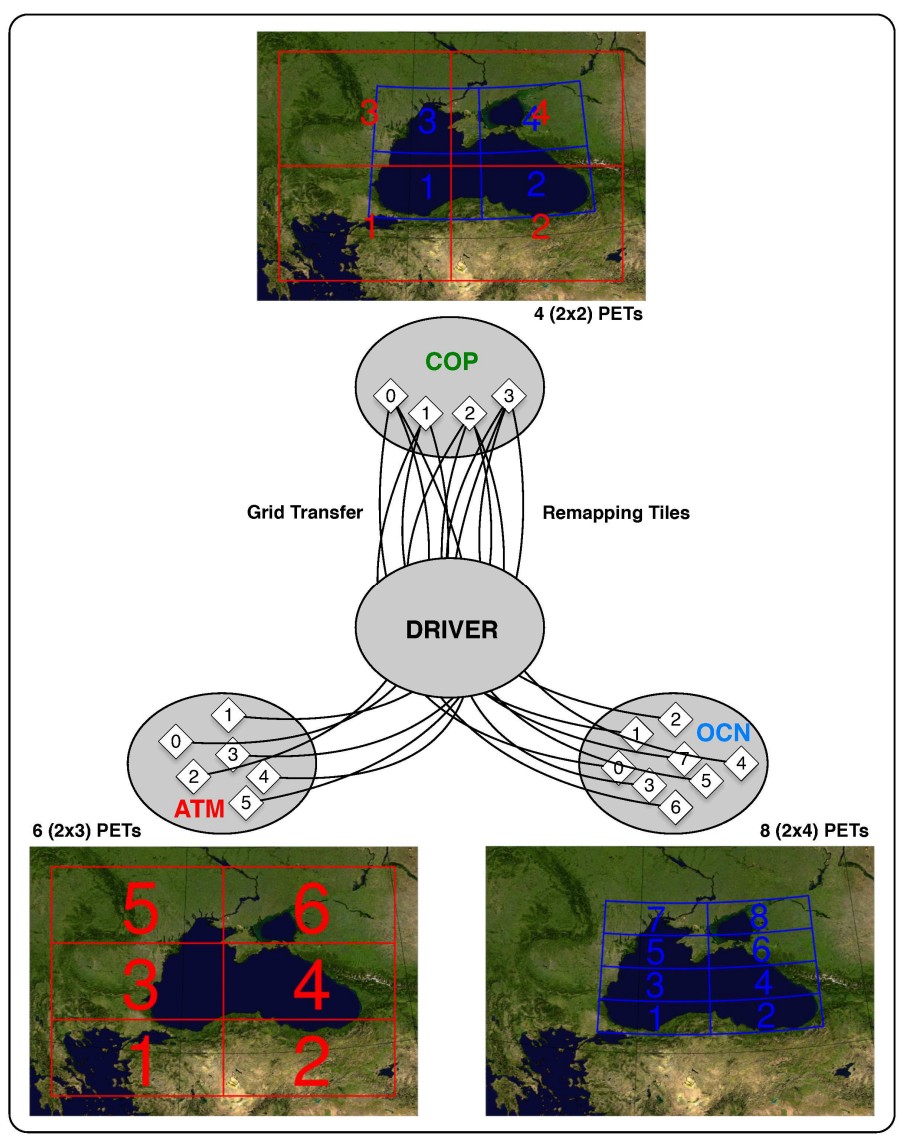

**Figure 7.** Two-component (atmosphere and ocean) representation of grid transfer and remapping feature of ESMF/NUOPC interface.

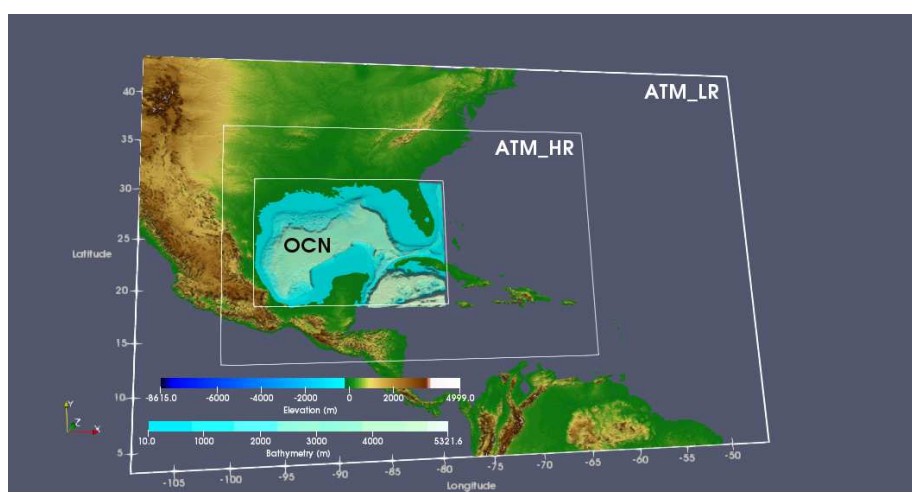

**Figure 8.** Domain for the RegESM simulations with topography and bathymetry of the region. The solid white boxes represent boundaries of the atmosphere (both outer and inner) and ocean model domains.

**Table 1.** Tested model configurations for benchmark simulations. Note that the dimension of vertical coordinates of ATM and OCN components are shown here after vertical interpolation from sigma to height and s-coordinates to depth. The visualization pipelines are also given as supplementary material.

| | P1: Case I | P2: Case II | P3: Case III |
|---|---|---|---|
| **Visualization** |  |  |  |
| **Pipeline** |  |  |  |
| **Primitives** | **ATM:** Contour for topography polyline for coastline and direct volume rendering for clouds | **ATM:** same with previous case but it includes iso-surface for wind speed and glyph for wind at specified level | **ATM:** Contour for topography, iso-surface for wind speed colored by relative humidity **OCN:** Contour for bathymetry, direct volume rendering for current |
| **Domain Size** | **ATM** **LR:** 170 x 235 x 27 **HR:** 880 x 1240 x 27 | **ATM** Same with **Case I** | **ATM** Same with **Case I** **OCN** 653 x 487 x 21 |
| **Number of Fields** | **1 x 3D ATM** Relative Humidity | **4 x 3D ATM** Relative Humidity Wind (u, v, w) | **4 x 3D ATM** Relative Humidity Wind (u, v, w) **4 x 3D OCN** Ocean Current (u, v, w) Land-Sea Mask |
| **Data Size ATM+OCN (MB)** | **LR:** 8.3 **HR:** 224.0 | **LR:** 33.2 **HR:** 896.0 | **LR:** 33.2+25.4 = 58.6 **HR:** 896.0+25.4 = 921.4 |
| **Time (s)** | **LR:** 2.3 – 3.7 **HR:** 17.7 – 65.0 | **LR:** 2.3 – 3.8 **HR:** 18.4 – 79.3 | **LR:** 6.8 – 14.6 **HR:** 7.8 – 10.1 |

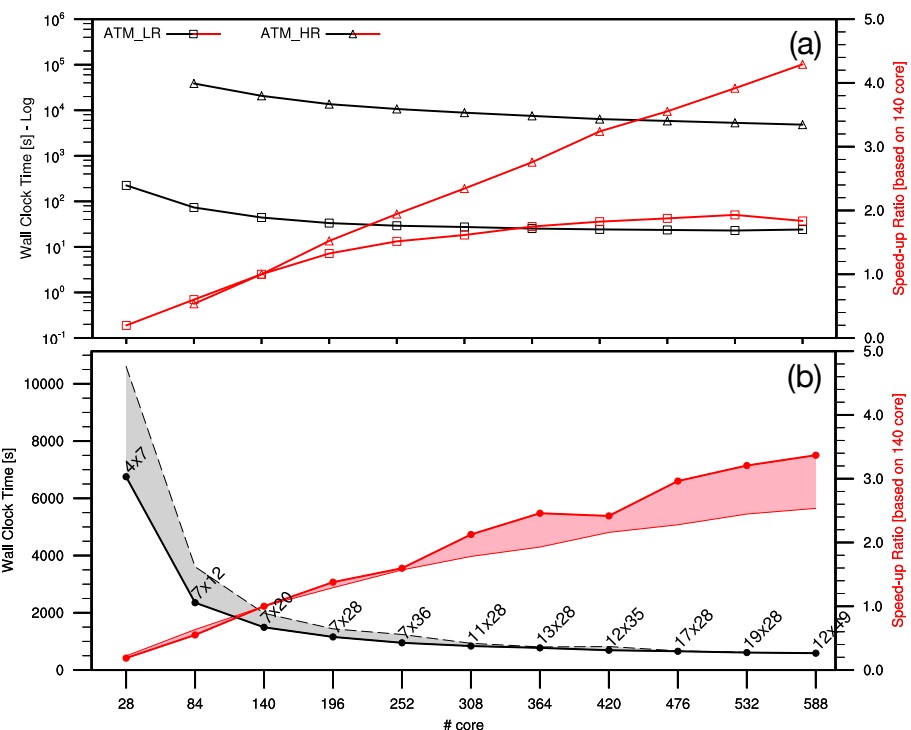

**Figure 9.** Benchmark results of standalone (a) atmosphere (ATM; both LR and HR) and (b) ocean (OCN) models. Note that timing results of the atmosphere model are in log axes to show both LR and HR cases in the same figure. The black lines represent measured wall clock times in second and red lines show speed-up. The envelope represents the timing and speed-up results that are done using the same number of cores but different two-dimensional decomposition configuration. The best two-dimensional decomposition parameters are also shown in the timing results for the ocean model case.

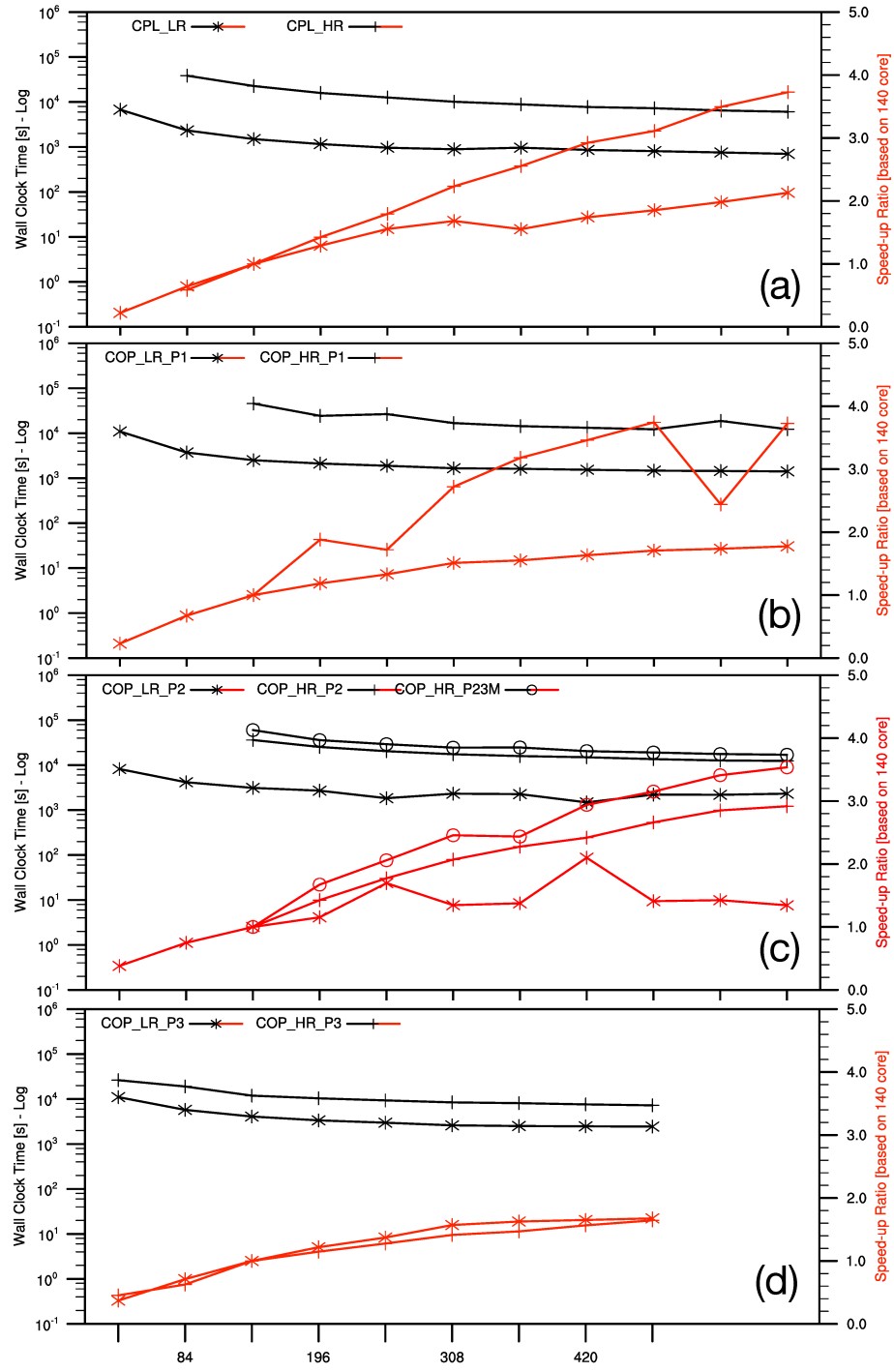

**Figure 10.** Benchmark results of (a) CPL simulations (b) COP simulations with P1 visualization pipeline, (c) COP simulations with P2 visualization pipeline and (d) COP simulations with P3 visualization pipeline. CPL represents the two-component modeling system (ATM and OCN), and COP indicates three-component modeling system (ATM, OCN and co-processing). Note that the HR case requires at least 140 cores to run and the speed-up results are given based on 140 cores.

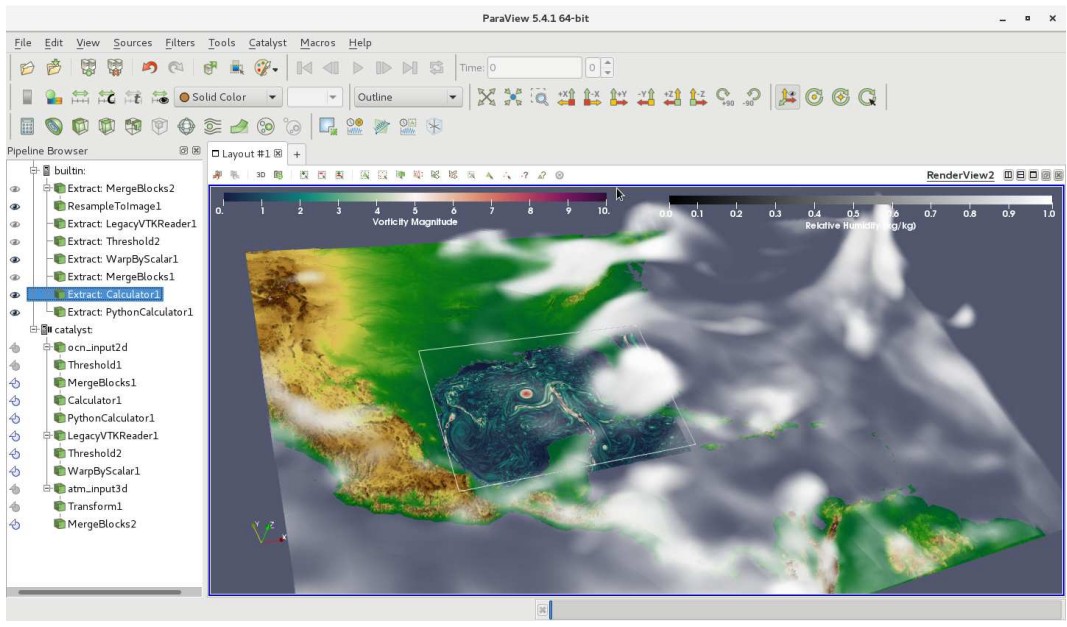

**Figure 11.** Volume rendering of atmospheric relative humidity field (*atm_input3d*) as well as vorticity field in the ocean surface (*ocn_input2d*) from COP_LR simulation using ParaView/Catalyst in live mode.

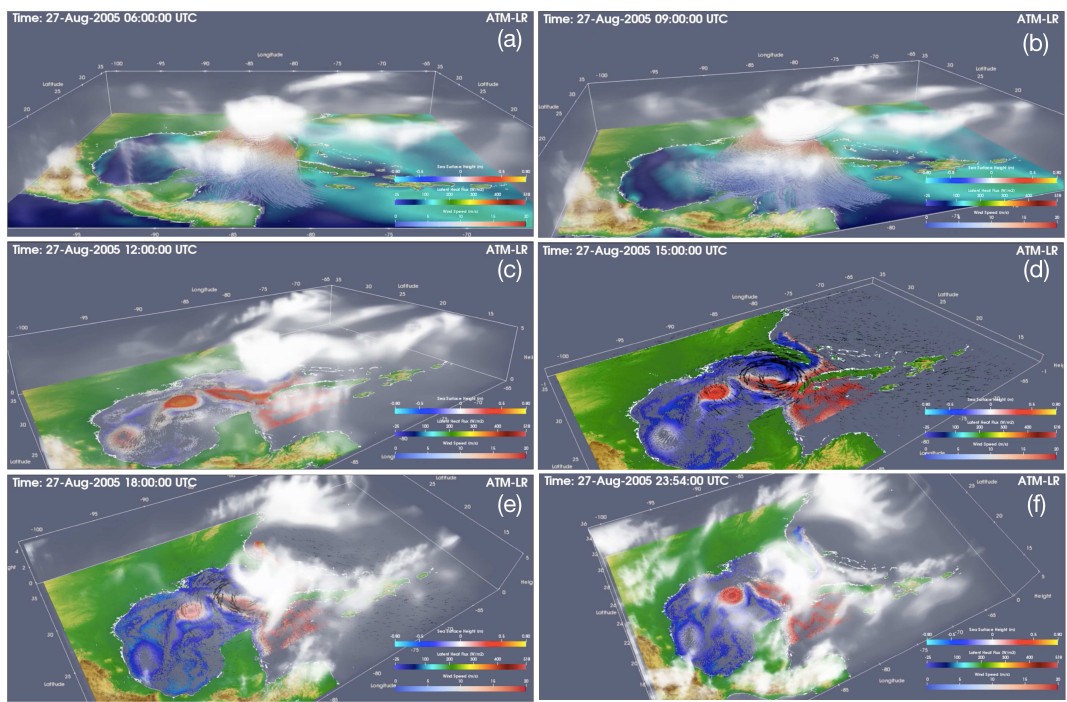

**Figure 12.** Rendering of multi-component (ATM-OCN-COP) fully coupled simulation using ParaView. The temporal interval for the processed data is defined as 6-minutes.

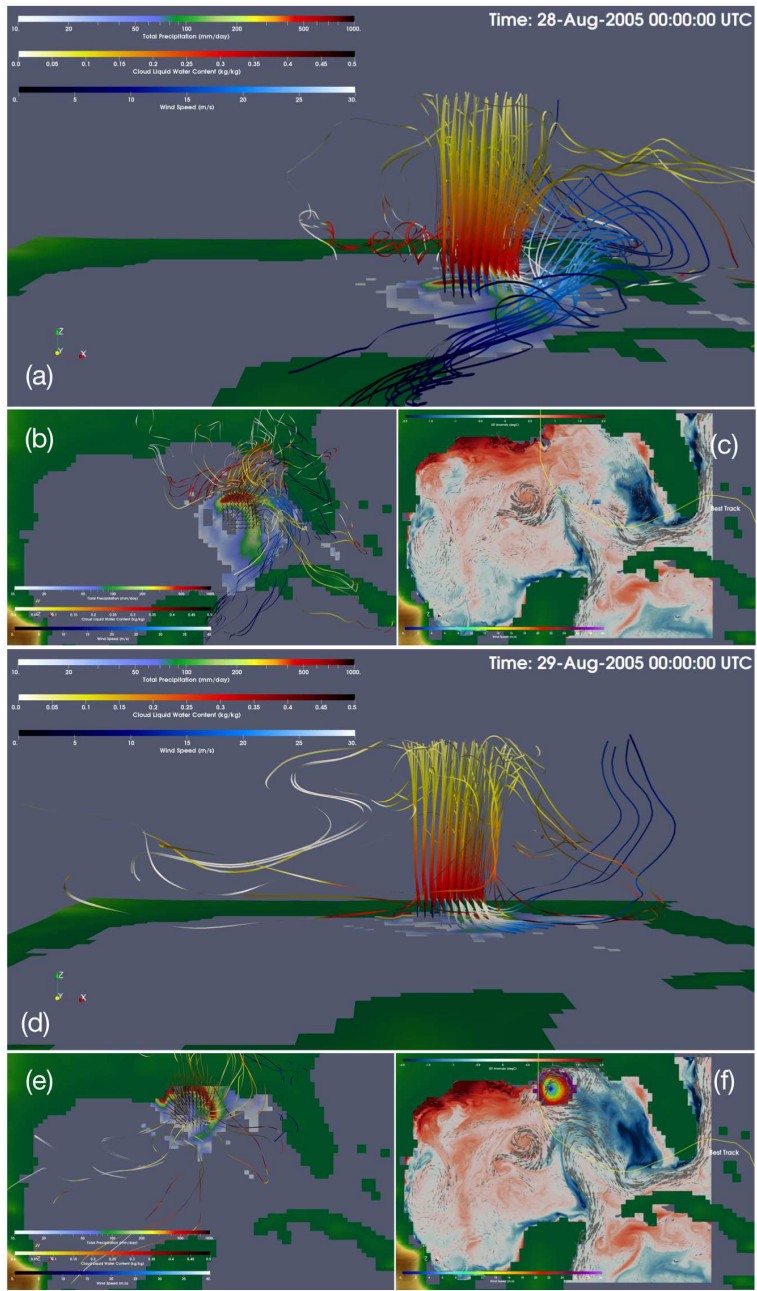

**Figure 13.** Rendering of three-dimensional vorticity streamlines ($s^{-1}$), total precipitation ($mmday^{-1}$) and sea surface temperature anomaly (degC) of COP_LR simulation for 28-Aug-2005 00:00 UTC **(a-c)** and 29-Aug-2005 00:00 UTC **(d-f)**. Streamlines are calculated only from the eye of the hurricane. In this case, red and yellow colored forward streamlines represent cloud liquid water content ($kgkg^{-1}$), and blue colored backward streamlines indicate wind speed ($ms^{-1}$). The solid yellow line represents the best track of Hurricane Katrina, which is extracted from HURDAT2 database. The larger versions of figures are also given as supplementary material.

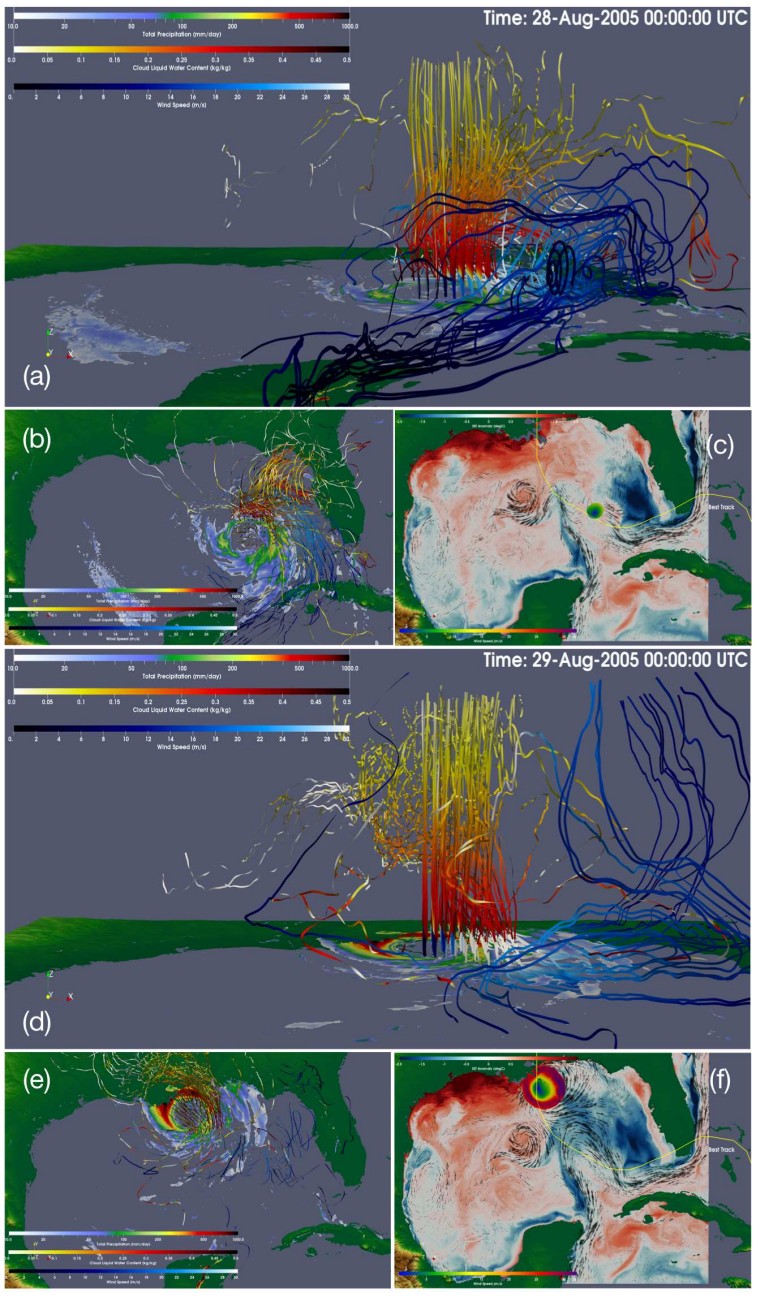

**Figure 14.** Same with Fig. 13 but for COP_HR simulation. The larger versions of figures are also given as supplementary material. The comparison of low and high-resolution model results is shown in the supplemental video.