# Peer review of "Toward modular in-situ visualization in Earth System Models: the regional modeling system RegESM 1.1"

_Geoscientific Model Development, 2018_

## Referee Comment (RC1) · C. Lemmen (Referee) · 5 Oct 2018

**General comments**

This single-author manuscript by Turuncoglu describes the scientific motivation, technical extension, and the application of the Regional Earth System Model (RegESM 1.0), originally also by the same author, with online visualization techniques. RegESM is originally a four-component modularly coupled system of ocean, wave, atmosphere and river routing science models. The convincingly argues that such coupled systems's growing data storage demands do not align with increases in compute efficiency, such

that storage of data for later analysis will necessarily have to compromise in how much or how often data can be stored. If data was stored at less time intervals, information on processes that are scientifically interesting and occur on very short time scales, could not be analysed with conventional tools, for example, storm events. By adding a modularly coupled visualization pipeline as part of the model system, events on short time scales could be recorded and condensed efficiently without the need to store the full model outputs at such short time scales. Consequently, Turuncoglu adds a fifth component to RegESM (now in version 1.1) that uses the widely used and parallelized visualization toolkit ParaView for recording fast time-scale data.

The contribution is within the scope of the Special Issue "Model infrastructure integration and interoperability (MI3)" in Geoscientific Model Development. The manuscript contributes to "combining multiple models into a coupled system", relates to "parallel data transfer and grid interpolation", and provides "supporting capabilities", here in the form of an already established visualization tool. The innovativity of the manuscript is the combination of the visualization with a modularly coupled system; this work has the potential to be of use to many other geoscientific model applications.

The paper is overall well written, but suffers from severe language issues pertaining to the use and misuse of articles, in particular, which inhibits the flow of reading. Wording, grammar, style and other aspects of language are mostly good, but this manuscript needs to be fixed by a native English speaker. Figures are appropriate with some potential for improvement to be more intelligible. The supplementary material is well presented and useful. The paper is well organized and has a clear and logical structure

I recommend that this paper is published after some scientific and substantial language revisions. Below I comment on specific aspects of the manuscript.

[Figure]

**Title, Abstract**

**title** Prepositions/articles missing. Also, I don't agree with "integrated" but suggest to use the word "modular". As the visualization has been proven, it is not a perspective "towards", but already production ready, so don't use "towards". I would rephrase the title (this is a mere suggestion) as "Co-processing online visualization: the Regional Earth System RegESM version 1.1".

**abstract** Fix language issues, such as: articles, avoid "being" before participles, check use of due to/because of, avoid overly use of "used", avoid "have to", do processes literally "move"? Avoid use of "basically", and many more. It is a major revision task to carefully check each sentence

**abstract** Avoid jargon especially in the abstract, it may not be clear to all interested readers what "multi-scale", "processing pipeline", "run process". Find alternative/easier wording or explain (as these terms are necessary and useful later in the manuscript). Define I/O (and consistently all abbreviations), define ESMF correctly.

**Introduction**

**p1 l23** Use consistent spelling for Earth System Model (capitalized), introduce abbreviation already here. Also use consistent spelling of Earth (name of our planet) versus earth.

**p2 l4** Unsubstantiated claims "more demanding compared to conventional standalone models, and in many cases, to global climate models". Also standalone models can be computationally demanding and though less complex may employ

a higher resolution in uncoupled contexts and lower resolution in coupled context, arriving at the same computational demand. And if global ESM were less demanding, also their resolution would increase aligned to available compute resources and user requirements.

**p2 l4** You have not yet defined "obvious needs". What are "truthful predictions" in a model context? Aren't they rather "accurate hindcasts" and "realistic projections".

**p2 l10f** avoid "in order", just use "to". Avoid "moreover" (throughout manuscript). Try to condense sentences and leave out all unnecessary words throughout the manuscript.

**p2 l12** "are also simplified" is not necessarily true. This "can" be simplified or is sometimes even "enforced", such as the grid specification in ESMF, or the standard names by CF conventions.

**p2 l16ff** ESMF suggests to use Theurich 2016 et al. for referencing ESMF. "in addition" does not align with "can be given as examples". MCT only provides part of the functionality that is present in EMSF (also FMS), as does OASIS (in its different); OASIS may or may not contain MCT, depending on version. There are also more recent developments, such as Jcup3, OMUSE, MOSSCO, MESSy, FABM, CSDMS which you may mention. Please clean up the list, discriminate couplers from coupling frameworks, sort by relationship and highlight key differences or commonalities. A review of older technologies is provided by Bert Jagers 2010.

**p2 l19** "The ESMs make use of state-of-art modeling frameworks, model components and libraries to have better representation of the earth system processes". This sentence is irrelevant. "The key component of coupled modeling systems is often the driver". Please refer to Alexander/Easterbrook (2015) for typical coupling constellations in ESMs. There, it is quite obvious that many models do not employ a driver. You could also use the term "hub-and-spoke" architecture (consistent

with your Figure 1) where the driver is the hub. The word "driver" itself is jargon and should be introduced beforehand.

**p2 l19** "such as six hourly or daily averages". This is true only for coupled physical systems. For geology ESMs, typical timescales may be longer (years, centuries), see e.g., CSDMS.

**p2 l23** "Coupling between individual earth system components adds extra overhead... network bandwidth usage, computation, disk I/O and data storage". Again, not necessarily. Please substantiate or give an example or reference where this applies. In my opinion the real overhead is in the loss of flexibility/modularity: it is preferable to have 1 hub speak a common language to 4 components than have each of the 4 components speak to each of the other three. Though the latter may be less complex, it hinders extensibility of a coupled system.

**p3 l3f** Fix articles. As everywhere.

**p3 l7** Don't overly use "in short", also throughout manuscript.

**p3 l7** Don't use judging language such as "brilliant". This entire sentence could be deleted (and the keyword "exascale" mentioned before.

**p4 l4ff** This paragraph is not needed.

**Design**

**p4 l9** "couple four different model components". I don't understand. Version 1.0 coupled four components, version 1.1 adds a co-processing component, totalling five. This is also confusing in Figure 1. Modular frameworks such as

NUOPC/ESMF do not discriminate scientific models, data components, output/visualization per se. All are just "components". I would suggest to use the term "component" for all five (four of them science models).

**p4 l11** delete "state-of-art"

**p4 l11** You have not yet introduced "the coupling interfaces" (this may be an article problem). So far, a driver and four models have been introduced. The necessity to add also coupling interfaces, and possibly NUOPC caps (CMI in the framework of Peckham 2013), has not yet been mentioned.

**p4 l16ff** delete this paragraph

**p4 l section heads** add ATM (later OCN, WAV, COP) to section heads

**p4 l22** For in-text enumerations, please use (1) with double parentheses, and do not write the number in bold face.

**p4 l26** "it is well tested with verified in". This does not make sense grammatically, please fix. Also, you have a verified model system for South and Eastern European/West Asian regions, but not for the Gulf of Mexico, which you provide as an example for the visualization.

**p4 l27f** Avoid "Additionally, ". What relevance has "does not support online nesting yet" for the application discussed here. Is online nesting needed for coupling to the co-processing model. Or is it just not "operational" in a sense that RegESM/WRF has not been scientifically proved?

**p5 sections 2.1.1/2.1.2** Align in model descriptions of RegCM and WRF the content. Use the same structure: dynamical core, spatial representation, explicitly modeled states and subgrid-scale parameterizations, sub-modules. This parallel constructions helps to compare the ATM science models.
**p5 l9** "the driver and coupling interface are developed by ITU". The driver should be independent of RegCM, why is this mentioned? What is the coupling interface? Is it the CMI/NUOPC cap? You may choose to introduce those two terms to make clear what you mean by coupling interface. Please describe the cap (e.g. what state variables does it export/import, which transformations does it encompass). You may also elaborate on the changes internal to RegCM (and WRF/MITgcm, WAM/HD)

**p5 l21** "model is modified to exchange data". Discriminate work done on cap (CMI) and within model (BMI, Peckham 2013).

**p5 l23ff** All that was said in the ATM section also applies here (parallel structure, elaboration ...)

**p6 l2ff** Avoid judging language like "fortunately". Avoid "in general" if possible. "In some studies", please reference those studies or give a motivation for both cases.

**p6 l7f** "In the current design of the coupled modeling system, the driver allows to select the desired exchange fields from the predefined list". Please explain how this works. Is the list like a database with all known fields from all possible coupled models and there is an automated selection of suitable fields?

**p6 l15** Leave out example technical detail "REGCM_COUPLING". I would appreciate a list of changes in the supplementary material, if that is practical to do.

**p6 l23** define PROTEUS abbreviation

**p6 l26** PBL has already been defined

**p6 l26f** third-generation *wave* model. What is "pure" physics? Use an en-dash to indicate a relationship in wave-wave / wave–wave (the second one is correct typography)

**p6/7 l32**  WAM is not by ENEA but by GKSS (now HZG) in Germany, I believe. Please attribute correctly. Describe changes to original model (here in text or as table in SOM).

**p7 l 5**  Unit of z0 (m).

**p7 l20f**  Explain your motivation for the two different ways. Do they pertain to ROMS and MITgcm coupling, respectively? How are they chosen?

**p7 l22f**  The change to output format is irrelevant in this context.

**p7 l30f**  "The ESMF is chosen because ... of the NUOPC". Please explain the relationship between ESMF and NUOPC. Why do you chose NUOPC in addition to ESMF, this needs elaboration for readers who are not familiar with NUOPC.

**p8 l2f**  "It also provides the capability of transferring computational grids"; here, "it" should refer to ESMF, not NUOPC.

**p8 l2f**  "NUOPC layer to support various configuration of component interactions such as defining multiple coupling time steps (fast and slow time steps; Fig. 2)". There are other frameworks that provide add this capability to ESMF, namely MAPL and MOSSCO, please reference and contrast.

**p8 l9f**  "connector components provided by NUOPC layer. Connector components are mainly used to create link between individual model components and driver". As far as I know, RegESM includes its own instance of a NUOPC connector, which is a single one for all connections between the components. Please elaborate on the design and functionality of the connector.

**p8 l18f**  You name stability as an advantage of semi-implicit coupling. Also add drawbacks of semi-implicit and advantages of implicit time stepping.

**p8 l20f**  delete "mainly" and "In this case"

**p8 l28** You may want to add the term "load balancing" for describing the challenges in concurrent coupling. There should be plenty of literature that describes load-balancing and possible ways to minimise this. In the geosciences, usually all OASIS-coupled models face this challenge.

**p8 l31f** The description of regridding could be a subsection, section 2.6 "Connector". How do other coupled model systems handle this (any references available?). Recently, I implemented into MOSSCO a very similar scheme using bilinear for interpolation and nearest neighbour for extrapolation, so I believe this is a very good way to handle this (but it is not published yet). Possibly, describe how ESMF 8 handles this rather than pointing vaguely to ongoing development.

**p9 l14** Align your description with the terms used in Figure 4 (see comments below on this figure).

**Integration of co-processing**

**p9 l25** Use an em-dash "—" instead of a hyphen. One sentence is not a paragraph, so join with next paragraph.

**p9 l28** Define "conventional" or give a reference to such approaches. Define pipeline in this context.

**p9 l30** How is "wrapper layer" different from Component Model Interface (CMI) or NUOPC cap or ESMF component interface?

**p10 l3** On what "other hand"? Describe the relationship between the adaptor and its NUOPC cap

**Benchmark**

**p14 l26** The "driver component introduces additional 5-10%". In ESMF documentations, usually below 5% overhead is assumed. this makes sense as you to the SMM twice for each interpolation of mask extrapolation. It would be interesting to see whether the generic implementation of extrapolation is more efficient in ESMF 8 (out of scope of this MS).

**p14 l35** " includes vertical interpolation to map data from sigma coordinates to height coordinates". Why is this interpolation step necessary? In principle, the visualization should equally work on sigma layers. And is the interpolation performed within the ESMF connector, or within a Paraview pipeline?

**p15 l2** the computational demand for visualization "require 10-40% extra" is rather high and its evaluation would probably change depending on whether it is more 10% (acceptable) or 40% (not acceptable). The benchmark using software rendering with Mesa is not ideal to demonstrate this, especially as many new HPC systems come with dedicated GPU nodes that could much reduce the visualization overhead.

**Results and Conclusions**

**p15 l31** This section is not well named. I suggest "Demonstration application" but make sure to choose something more elaborate than "results"

**p15 l35** oblivious => obvious

**p16 l3** Explain how you choose between live mode and co-processing modes. Are these two different models? Or is this done by way of a configuration file. Or,

is it possible to attach live viewing to any running system that has co-processing enabled?

**Figures**

**Figure 1**  Very clear and readable, consistent color and design with NUOPC web page. Please see my suggestions to treat everything as component, regardless of science "MODEL" or visualization "COMP". It is not clear from the text what "redist" is; also, this feature is not used when running sequentially. I also do not understand the many "regrid" arrows, as regridding occurs when the driver transports a field from one to another component, not from the component to the driver. This is confusing.

**Figure 4**  Rather unreadable. Please redesign. (1) Avoid bridges (this is possible by routing the arrow "Create ESMF FIELD_B" north to "Create ESMF FIELD_C". (2) don't use same line thickness/color for outlining the boxes around src/dst grid, rather make them visually less intrusive to highlight the flowchart. (3) Abstract away 1.0, LARGEST, "over sea" or other value-based masking, if possible.

**Figure 7**  It is confusing that the visualization 2x2 tiles don't geographically overlap for data from OCN and ATM in CPL. This is not needed, but confusing at first. Maybe give a hint to the reader that this is not a problem.

**Figures 9/10**  It is very difficult to compare ATM versus OCN performance using different y scales. I recommend to use the same ylog-scale for all panels in figs 9 and 10. Also try loglog. Tile sizes are hard to read, please move text from lines. Calculate all speedup relative to 140 cores, so that you don't have different reference points. Explain model versions like RegCM_r6274, delete UHeM. Avoid overlapping graphics and text.
**Figure 10** 8 lines in a graph is too messy. Even if on ylog, this may be difficult. Possibly choose different or more subpanels (e.g. CPL_LR + CPL_HR in one panel, then P1, P2, P3 each in one panel). Or use more colors for lines or different line styles ...

**Figure 11** Zoom in to Paraview Window only (delete lower and upper OS information bands)

**Figure 13** Use kg kg$^{-1}$ and the like, not the slash for division in units.

**1 Code**

I have been able to download and compile all components, except for getting the catalyst to run. It would be helpful to provide a fully functional reference system as a docker container, for example.

**2 Supplementary videos**

Useful and nice! Please make sure the video dois are referenced in the text, and add to the videos a link to your publication.

---

## Referee Comment (RC2) · R. Dunlap (Referee) · 5 Oct 2018

This paper describes a novel approach to in-situ (online) analysis and visualization of numerical model output by integrating the visualization package with a model coupling framework, the Earth System Modeling Framework (ESMF). Key contributions of the work include the ability to analyze fast moving processes at higher temporal and spatial resolution than would typically be possible (due to extreme size of data output) as well as offering a generic, reusable approach that could be applied to other models using ESMF.

The paper is well-written and clearly describes limitations of current data analysis ap-

proaches and how the proposed architecture with integrated in-situ visualization addresses them. Existing approaches to in-situ visualization are discussed including implementations using the Model for Prediction Across Scales (MPAS) and Weather Research and Forecasting (WRF) models. However, these approaches use custom implementations, do not leverage standardized coupling interfaces, and are therefore could be hard to apply generally across a range of models.

The paper describes the architecture of the RegESM model, which supports coupled atmosphere, ocean, wave, and river models. The driver and coupling protocols are based on the National Unified Operational Prediction Capability (NUOPC) software layer. This architecture was extended to include the ParaView/Catalyst co-processing component using the same ESMF-based data structures and parallel communication operations used to exchange coupling data between the model components.

A major strength of the paper is that the fully integrated system was tested using real model components performing a simulation of Hurricane Katrina. This allowed for analysis of the hurricane at very high temporal resolution. Timing profiles of the full system show reasonable scaling for two separate resolutions, up to 588 cores.

Since a key focus on the paper is interoperability afforded by using a standard coupling framework, some additional discussion on details of the software engineering and approach to interoperability could be discussed including more details on the actual interfaces used between components as well as issues related to portability. Follow on work could look at applying the same co-processing component to a completely different model to understand how generic the approach is. A related question is how hard it would be to change out the visualization package itself, since there are a number of packages that offer custom analyses.

In addition to timing profiles and since the initial motivation was around the problem of data volumes, the paper could benefit from plots describing the amount of data exchanged and used in the in-situ case versus the amount of data that would be required

for offline visualization at the same temporal frequency. This would allow an "apples-to-apples" comparison of the online vs. offline approaches.

The overall approach is novel and represents a significant contribution to the community, especially given the large number of models using the ESMF/NUOPC framework. If the work is extended, many of these models could benefit from the in-situ visualization approach.

---

## Author Comment (AC1) · 2 Nov 2018

**Response to Reviewer I This document is color-coded as follows:**

- Comments by reviewers are in **blue**.
- Our responses are in **black**.
- Blocks of text that is added to the are in **red**
- **The track-changes can be also seen as a supplement document**

**Title, Abstract**

**1)** title Prepositions/articles missing. Also, I don't agree with "integrated" but suggest to use the word "modular". As the visualization has been proven, it is not a perspective "towards", but already production ready, so don't use "towards". I would rephrase the title (this is a mere suggestion) as "Co-processing online visualization: the Regional Earth System RegESM version 1.1".

In the current implementation, the adaptor code between NUOPC cap and ParaView, Catalyst is just a prototype that supports only for regional modelling cases with limited spatial grid support (structured and curvilinear grids). It still needs additional work to be more generic and adaptor code needs to be extended to include support for different mesh types such as unstructured grid for both global and regional applications. These are not implemented yet and needs to design adaptor code carefully for more generic applications. This will be done in the future.

**2)** abstract Fix language issues, such as: articles, avoid "being" before participles, check use of due to/because of, avoid overly use of "used", avoid "have to", do processes literally "move"? Avoid use of "basically", and many more. It is a major revision task to carefully check each sentence

Abstract section is modified based on your suggestions.

**3)** abstract Avoid jargon especially in the abstract, it may not be clear to all interested readers what "multi-scale", "processing pipeline", "run process". Find alternative/easier wording or explain (as these terms are necessary and useful later in the manuscript). Define I/O (and consistently all abbreviations), define ESMF correctly.

Abstract section is modified and simplified by removing most of the domain specific jargons.

**Introduction**

**4)** p1 l23 Use consistent spelling for Earth System Model (capitalized), introduce abbreviation already here. Also use consistent spelling of Earth (name of our planet) versus earth.

The text is modified based on your suggestions.

**5)** p2 l4 Unsubstantiated claims "more demanding compared to conventional standalone models, and in many cases, to global climate models". Also standalone models can be computationally demanding and though less complex may employ a higher resolution in uncoupled contexts and lower resolution in coupled context, arriving at the same computational demand. And if global ESM were less demanding, also their resolution would increase aligned to available compute re- sources and user requirements.

The sentence is removed.

**6)** p2 l4 You have not yet defined "obvious needs". What are "truthful predictions" in a model context? Aren't they rather "accurate hindcasts" and "realistic projections".

The sentence is modified based on your suggestions and moved to the end of the paragraph.

**7)** p2 l10f avoid "in order", just use "to". Avoid "moreover" (throughout manuscript). Try to condense sentences and leave out all unnecessary words throughout the manuscript.

The sentence is modified based on your suggestions.

**8)** p2 l12 "are also simplified" is not necessarily true. This "can" be simplified or is sometimes even "enforced", such as the grid specification in ESMF, or the standard names by CF conventions.

The sentence is modified based on your suggestions.

**9)** p2 l16ff ESMF suggests to use Theurich 2016 et al. for referencing ESMF. "in addition" does not align with "can be given as examples". MCT only provides part of the functionality that is present in EMSF (also FMS), as does OASIS (in its different); OASIS may or may not contain MCT, depending on version. There are also more recent developments, such as Jcup3, OMUSE, MOSSCO, MESSy, FABM, CSDMS which you may mention. Please clean up the list, discriminate couplers from coupling frameworks, sort by relationship and highlight key differences or commonalities. A review of older technologies is provided by Bert Jagers 2010.

The more recent reference of ESMF is added to the list. The literature of couplers and coupling frameworks are also extended. Please refer to the supplementary document that includes track-changes.

**10)** p2 l19 "The ESMs make use of state-of-art modeling frameworks, model components and libraries to have better representation of the earth system processes". This sentence is irrelevant. "The key component of coupled modeling systems is often the driver". Please refer to Alexander/Easterbrook (2015) for typical coupling constellations in ESMs. There, it is quite obvious that many models do not employ a driver. You could also use the term "hub-and-spoke" architecture (consistent with your Figure 1) where the driver is the hub. The word "driver" itself is jargon and should be introduced beforehand.

The paragraph is also modified to refer Alexander/Easterbrook (2015) paper that briefly demonstrate the diversity of the current global earth system models.

**11)** p2 l19 "such as six hourly or daily averages". This is true only for coupled physical systems. For geology ESMs, typical timescales may be longer (years, centuries), see e.g., CSDMS.

It is removed from the sentence.

**12)** p2 l23 "Coupling between individual earth system components adds extra overhead... network bandwidth usage, computation, disk I/O and data storage". Again, not necessarily. Please substantiate or give an example or reference where this applies. In my opinion the real overhead is in the loss of flexibility/modularity: it is preferable to have 1 hub speak a common language to 4 components than have each of the 4 components speak to each of the other three. Though the latter may be less complex, it hinders extensibility of a coupled system.

The misleading sentence is removed from manuscript.

**13)** p3 l3f Fix articles. As everywhere.

I am aware that there are numerous missing articles in the manuscript and these are fixed in the revised version of manuscript.

**14)** p3 l7 Don't overly use "in short", also throughout manuscript.

It is modified as "as a result" in here.

**15)** p3 l7 Don't use judging language such as "brilliant". This entire sentence could be deleted (and the keyword "exascale" mentioned before.

It is changed to "state-of-art" and "exascale" is deleted.

**16)** p4 l4ff This paragraph is not needed.

It is removed.

**Design**

**17)** p4 l9 "couple four different model components". I don't understand. Version 1.0 coupled four components, version 1.1 adds a co-processing component, totalling five. This is also confusing in Figure 1. Modular frameworks such as NUOPC/ESMF do not discriminate scientific models, data components, out-put/visualization per se. All are just "components". I would suggest to use the term "component" for all five (four of them science models).

The paragraph is modified based on your suggestions.

**18)** p4 l11 delete "state-of-art"

It is removed.

**19)** p4 l11 You have not yet introduced "the coupling interfaces" (this may be an article problem). So far, a driver and four models have been introduced. The necessity to add also coupling interfaces, and possibly NUOPC caps (CMI in the framework of Peckham 2013), has not yet been mentioned.

The coupling interface is introduced before its use and hub-and-spoke architecture is also mentioned in here.

**20)** p4 l16ff delete this paragraph

It is removed.

**21)** p4 l section heads add ATM (later OCN, WAV, COP) to section heads

They are added

**22)** p4 l22 For in-text enumerations, please use (1) with double parentheses, and do not write the number in bold face.

It is fixed. Same convention is also used in everywhere in the manuscript.

**23)** p4 l26 "it is well tested with verified in". This does not make sense grammatically, please fix. Also, you have a verified model system for South and Eastern European/West Asian regions, but not for the Gulf of Mexico, which you provide as an example for the visualization.

It is fixed. The RegCM atmospheric model component is used because WRF is not ready to couple with co-processing component. I still need to add three-dimensional fields such as mixing ratio etc. along with support of vertical interpolation. We are currently working on it as a part of ongoing national science project and will be ready soon (in version 1.2).

**24)** p4 l27f Avoid "Additionally, ". What relevance has "does not support online nesting yet" for the application discussed here. Is online nesting needed for coupling to the co-processing model. Or is it just not "operational" in a sense that RegESM/WRF has not been scientifically proved?

It is fixed. No, there is no need to use on-line nesting to couple with co-processing. It is also not supported in the current version of the coupled modeling system. So, it is removed. We are planning to redesign whole modeling system from scratch to support online nesting among model components but this is our long-term goal for RegESM 2.0. We are also having fully coupled WRF-MITgcm model for Black Sea but the results are not published yet. Along with a help of newly introduced co-processing

component, the Black Sea model will be used to investigate lake-snow effect as a part of same ongoing national project.

**25)** p5 sections 2.1.1/2.1.2 Align in model descriptions of RegCM and WRF the content. Use the same structure: dynamical core, spatial representation, explicitly modeled states and subgrid-scale parameterizations, sub-modules. This parallel construction helps to compare the ATM science models.

Both sections are restructured to align their content as much as possible. In this case, they have very brief information about models and reader must refer to their own documents to get more information.

**26)** p5 l9 "the driver and coupling interface are developed by ITU". The driver should be independent of RegCM, why is this mentioned? What is the coupling interface? Is it the CMI/NUOPC cap? You may choose to introduce those two terms to make clear what you mean by coupling interface. Please describe the cap (e.g. what state variables does it export/import, which transformations does it encompass). You may also elaborate on the changes internal to RegCM (and WRF/MITgcm, WAM/HD)

The sentence that refers to ITU and ENEA is removed. The processes handled by NUOPC cap are listed in here because they are same for both atmospheric model component. The NUOPC cap and details about the implementation are added.

**27)** p5 l21 "model is modified to exchange data". Discriminate work done on cap (CMI) and within model (BMI, Peckham 2013).

The text is modified and now the distinction between modifications done in model side and cap is more apparent. In each model section, only work done in model side is described.

**28)** p5 l23ff All that was said in the ATM section also applies here (parallel structure, elab- oration ...)

Both sections are restructured to align their content as much as possible.

**29)** p6 l2ff Avoid judging language like "fortunately". Avoid "in general" if possible. "In some studies", please reference those studies or give a motivation for both cases.

 "fortunately" is removed from the manuscript. "In general" is also removed and i refered to our previous work for the bulk flux algorithm.

Turuncoglu, U. U., Giuliani, G., Elguindi, N., and Giorgi, F.: Modelling the Caspian Sea and its catchment area using a coupled regional atmosphere-ocean model (RegCM4-ROMS): model design and preliminary results, Geosci. Model Dev., 6, 283-299, https://doi.org/10.5194/gmd-6-283-2013, 2013.

In this study, we have to use bulk flux algorithm because ice model coupled with ROMS (not latest CICE version of ROMS, Budgell, 2005 version is used) was not working with the direct fluxes provided by atmospheric model.

**30)** p6 l7f "In the current design of the coupled modeling system, the driver allows to select the desired exchange fields from the predefined list". Please explain how this works. Is the list like a database with all known fields from all possible coupled models and there is an automated selection of suitable fields?

It is just a list of fields that component could import and export. The automatization could be implemented in future version of the modeling system by using ontologies, common vocabularies and conventions.

**31)** p6 l15 Leave out example technical detail "REGCM_COUPLING". I would appreciate a list of changes in the supplementary material, if that is practical to do.

It is removed. The documentation of the RegESM modeling system includes all those tiny details. The reader could refer to the following link https://github.com/uturuncoglu/RegESM

**32)** p6 l23 define PROTEUS abbreviation

Actually, I searched in web and their publications but i could not find the source of the abbreviation but ENEA is using it for their publications and web site. So, I prefer to keep it as it is.

**33)** p6 l26 PBL has already been defined

It is corrected.

**34)** p6 l26f third-generation wave model. What is "pure" physics? Use an en-dash to indicate a relationship in wave-wave / wave–wave (the second one is correct typography)

"pure physcis" is removed. wave-wave is also written with en-dash now.

**35)** p6/7 l32 WAM is not by ENEA but by GKSS (now HZG) in Germany, I believe. Please attribute correctly. Describe changes to original model (here in text or as table in SOM).

It is fixed now. The list of changes in WAV side is also included.

**36)** p7l5 Unit of z0(m).

It is added.

**37)** p7 l20f Explain your motivation for the two different ways. Do they pertain to ROMS and MITgcm coupling, respectively? How are they chosen?

The first option is used to define river plumes correctly while the second one is generally used to distribute river discharge to a large areal extent over the ocean. These are defined in the configuration file of the driver. The section is also modified to include this information.

**38)** p7 l22f The change to output format is irrelevant in this context.

It is removed.

**39)** p7 l30f "The ESMF is chosen because ... of the NUOPC". Please explain the relationship between ESMF and NUOPC. Why do you chose NUOPC in addition to ESMF, this needs elaboration for readers who are not familiar with NUOPC.

The section is extended to include more information about relationship between ESMF and NUOPC interface.

**40)** p8 l2f "It also provides the capability of transferring computational grids"; here, "it" should refer to ESMF, not NUOPC.

It is fixed.

**41)** p8 l2f "NUOPC layer to support various configuration of component interactions such as defining multiple coupling time steps (fast and slow time steps; Fig. 2)". There are other frameworks that provide add this capability to ESMF, namely MAPL and MOSSCO, please reference and contrast.

Actually, this feature is mainly inherited from ESMF by MAPL and MOSSCO. I think it is better to reference to ESMF/NUOPC for it.

**42)** p8 l9f "connector components provided by NUOPC layer. Connector components are mainly used to create link between individual model components and driver". As far as I know, RegESM includes its

own instance of a NUOPC connector, which is a single one for all connections between the components. Please elaborate on the design and functionality of the connector.

There is no single connector in the modeling system. For example, when RegESM is configured to couple only atmosphere and ocean components. The modelling system creates two connectors to represent ATM-OCN and OCN-ATM interactions. This is little bit different when atmosphere model coupled with co-processing component. In this case, there is only one connector because only ATM-COP interaction is valid. COP component does not send any information to other components (like a sink). The section is extended to include this information.

**43)** p8 l18f You name stability as an advantage of semi-implicit coupling. Also add draw-backs of semi-implicit and advantages of implicit time stepping.

The paragraph is extended to include more information about different coupling schemes.

**44)** p8 l20f delete "mainly" and "In this case"

They are removed.

**45)** p8 l28 You may want to add the term "load balancing" for describing the challenges in concurrent coupling. There should be plenty of literature that describes load- balancing and possible ways to minimise this. In the geosciences, usually all OASIS-coupled models face this challenge.

Yes, the right term is load-balancing. A reference is added about LUCIA (Load-balancing Utility and Coupling Implementation Appraisal) tool for load-balancing analysis in OASIS-MCT based coupled systems.

**46)** p8 l31f The description of regridding could be a subsection, section 2.6 "Connector". How do other coupled model systems handle this (any references available?). Recently, I implemented into MOSSCO a very similar scheme using bilinear for interpolation and nearest neighbour for extrapolation, so I believe this is a very good way to handle this (but it is not published yet). Possibly, describe how ESMF 8 handles this rather than pointing vaguely to ongoing development.

I prefer to keep it as it is. The special interpolation implemented in RegESM modeling system is a part of driver.

**47)** p9 l14 Align your description with the terms used in Figure 4 (see comments below on this figure).

We used field names (Field_A, Field_B and Field_C) in also manuscript. The paragraph is also extended to describe the algorithm clearly.

**Integration of co-processing**

**48)** p9 l25 Use an em-dash "—" instead of a hyphen. One sentence is not a paragraph, so join with next paragraph.

It is fixed.

**49)** p9 l28 Define "conventional" or give a reference to such approaches. Define pipeline in this context.

The conventional co-processing enabled simulation systems interacts with single physical model component such as atmosphere along with co-processing support. This is defined also in text and references are added.

… A visualization pipeline integrates a data flow network in which computation is described as a collection of executable modules that are connected in a directed graph representing how data moves between modules (Moreland, 2013). There are three types of modules: sources (file readers and

**50)** p9 l30 How is "wrapper layer" different from Component Model Interface (CMI) or NUOPC cap or ESMF component interface?

A NUOPC cap is defined for co-processing component along with adaptor code. In this case, adaptor code is an interface for ParaView, Catalyst API and NUOPC cap passes information from ESMF to adaptor code. The text slightly modified to clear the design.

**51)** p10 l3 On what "other hand"? Describe the relationship between the adaptor and its NUOPC cap

It is removed. It is defined in previous paragraph.

**Benchmark**

**52)** p14 l26 The "driver component introduces additional 5-10%". In ESMF documentations, usually below 5% overhead is assumed. this makes sense as you to the SMM twice for each interpolation of mask extrapolation. It would be interesting to see whether the generic implementation of extrapolation is more efficient in ESMF 8 (out of scope of this MS).

I agree. As far as i know, the ESMF developers was trying to find a way to combine weight matrix of multiple interpolation (including extrapolation) into a single one. Similar methodology is using in transformation (scale, rotation, translation etc.) matrixes using homogeneous coordinates but i am not sure it could be used in regridding or not.

**53)** p14 l35 " includes vertical interpolation to map data from sigma coordinates to height coordinates". Why is this interpolation step necessary? In principle, the visualization should equally work on sigma layers. And is the interpolation performed within the ESMF connector, or within a Paraview pipeline?

I did not try yet but yes the visualization works with sigma coordinates without any problem. The vertical interpolation is introduced to have a consistency in the data coming from atmosphere and ocean. By this way, the vertical scales of the data can be compared without any problem. In general, both atmosphere and ocean components use sigma coordinates with different definitions. The interpolation is done in ESMF cap not in ParaView side. It could be done in ParaView side also but in this case all grid related parameters such as stretching functions etc. need to be passed to ParaView, Catalyst and special ProgrammableFilter need to be developed.

**54)** p15 l2 the computational demand for visualization "require 10-40% extra" is rather high and its evaluation would probably change depending on whether it is more 10% (acceptable) or 40% (not acceptable). The benchmark using software rendering with Mesa is not ideal to demonstrate this, especially as many new HPC systems come with dedicated GPU nodes that could much reduce the visualization overhead.

It is clear that Mesa is not right way to measure the performance of the co-processing component. The overall performance of the used computing system dominates the results in our case but we don't have accesses to a system with GPU accelerators. In the early stage of the performance benchmark, Dell provided us a grant in newly installed University Manchester system (10000 core/hour) but the system is mainly designed for GPU intensive and AI type researches and the nodes have small number of core (10 cores) but four NVIDIA P100 GPU. In fact, our test simulations are CPU intensive and requires high amount of compute resource (or cores) for the high-resolution simulations and rendering only done on a single GPU (limitation of ParaView, Catalyst and used filters). As a result, couple of benchmark run spend all allocated resources (unused GPU resources was also accounted) and the system was not used for the benchmark. There was also issue related with buggy NVIDIA driver and ParaView protobuf library. In the future, along with the development of m-VTK (GPU accelerated filters) it could be possible to use multi-GPU systems.

**Results and Conclusions**

**55)** p15 l31 This section is not well named. I suggest "Demonstration application" but make sure to choose something more elaborate than "results"

It is changed as suggested.

**56)** p15 l35 oblivious => obvious

It is fixed.

**57)** p16 l3 Explain how you choose between live mode and co-processing modes. Are these two different models? Or is this done by way of a configuration file. Or, is it possible to attach live viewing to any running system that has co-processing enabled?

No, it is not a different model or configuration. The selection of the mode is controlled by the Python script generated using ParaView, co-processing plugin. The user could also activate live visualization mode, just by changing a single line of code (coprocessor.EnableLiveVisualization needs to be set as True) in the co-processing Python script. Yes, if live visualization is enabled, the user can attach to the simulation anytime and pause it to analyze the results. This information is added to section.

**Figures**

**58)** Figure 1 Very clear and readable, consistent color and design with NUOPC web page. Please see my suggestions to treat everything as component, regardless of sci- ence "MODEL" or visualization "COMP". It is not clear from the text what "redist" is; also, this feature is not used when running sequentially. I also do not under- stand the many "regrid" arrows, as regridding occurs when the driver transports a field from one to another component, not from the component to the driver. This is confusing.

The figure is modified. All the boxes are named as "COMPONENT" and the texts are also removed from the arrows.

**59)** Figure 4 Rather unreadable. Please redesign. (1) Avoid bridges (this is possible by routing the arrow "Create ESMF FIELD_B" north to "Create ESMF FIELD_C". (2) don't use same line thickness/color for outlining the boxes around src/dst grid, rather make them visually less intrusive to highlight the flowchart. (3) Abstract away 1.0, LARGEST, "over sea" or other value-based masking, if possible.

The figure is modified. The hop is solved by changing the position of the arrow. The dashed line is used in the boxes and color is changed to grey to move focus to workflow. It is hard to abstract used values and mask values. The algorithm is tested with this configuration and it is better to keep them as it is to have a reproducible result (at least for finding mapped and unmapped grid points). I am also using same methodology using NCL ESMF interface to create forcing for ocean models and it works better than standard extrapolation method (based on solution of Possion Eq.) provided by NCL itself.

**60)** Figure 7 It is confusing that the visualization 2x2 tiles don't geographically overlap for data from OCN and ATM in CPL. This is not needed, but confusing at first. Maybe give a hint to the reader that this is not a problem.

The order of plots and labels of Fig 6 and Fig 7 was wrong. It is fixed now. Following section is also included to the manuscript

… In this case, ATM and OCN model components do not need to geographically overlap for co-processing. The only limitation is that the ATM component must cover the entire OCN model domain for an ATM-OCN coupled system because the ATM component provides the surface boundary condition for OCN component. …

**61)** Figures 9/10 It is very difficult to compare ATM versus OCN performance using different y scales. I recommend to use the same ylog-scale for all panels in figs 9 and 10. Also try loglog. Tile sizes are hard to read, please move text from lines. Calculate all speedup relative to 140 cores, so that you don't have different reference points. Explain model versions like RegCM_r6274, delete UHeM. Avoid overlapping graphics and text.

The speed-up calculation of coupled model simulations starts from 140 core because this is the minimum number of cores that allows to run the modeling system. For standalone simulations, it is possible to run the model with 28 cores (single node). As a result, it was used 28 cores for standalone and 140 cores for coupled model simulations to calculate the speedup. The plots are modified to have consistent x and y axis (log scale) as well as number of cores used in the speed-up calculation (140 cores) except plot for standalone ocean model. In this case, using same log scale for it does not give good results as expected for example the effect of tile configuration cannot be seen clearly because the benchmark results fall into a very narrow interval. To that end, i prefer to keep results of standalone ocean simulations as it is. I also increased font sizes and move the text little bit far from the markers for standalone ocean model plot. The texts that indicate model version and name of the computing system that is used in the benchmark are removed because they are already mentioned in the manuscript.

**62)** Figure 10 8 lines in a graph is too messy. Even if on ylog, this may be difficult. Possibly choose different or more subpanels (e.g. CPL_LR + CPL_HR in one panel, then P1, P2, P3 each in one panel). Or use more colors for lines or different line styles ...

The plot is modified. Now it has four panel. The first one (a) compares coupled model simulations (w/o co-processing component), and the rest compares coupled model simulations with co-processing component enabled (LR vs. HR). (b) for P1 visualization pipelines, (c) for P2 visualization pipelines and (d) for P3 visualization pipelines. The both y scales (the left one uses log scale) are also consistent now.

**63)** Figure 11 Zoom in to Paraview Window only (delete lower and upper OS information bands)

It is fixed.

**64)** Figure 13 Use kg kg−1 and the like, not the slash for division in units.

The figure label is fixed.

**Code**

**65)** I have been able to download and compile all components, except for getting the catalyst to run. It would be helpful to provide a fully functional reference system as a docker container, for example.

The Docker container was used to do online demo of the modeling system in the last NVIDIA GTC (GPU Technology Conference). More information about demo case and created Docker container (using software rendering) can be found in following public repository (DOI is also created by Zenodo and referenced in code availability section).

https://github.com/uturuncoglu/GTC2018_demo

There is also Docker container (NVIDIA docker need to be installed - https://github.com/NVIDIA/nvidia-docker) that supports NVIDIA GPUs and can be found in following link

https://drive.google.com/file/d/1hz0Frbawm2UxNtSBnjIrGSHlqDk1kZFY/view?usp=sharing

In this case, the file size is greater than upper limit of GitHub LFS (Large File Support; 2 GB) and it is shared with Google Drive. After getting file following commands can be used to run the Docker container and low-resolution version of coupled model with co-processing support.

**1)** import container
docker import nvidia_opengl_gtc2018-egl.tar.gz
**2)** list images to retrieve IMAGE ID
docker images
**3)** tag IMAGE (52ebebd92125 is just an example for Image ID, use your own from previous command)
docker tag 52ebebd92125 nvidia/opengl:gtc2018-egl
**4)** run container and get interactive shell
nvidia-docker run -it nvidia/opengl:gtc2018-egl /bin/bash
**5)** go to simulation directory (all required libraries installed under /opt/progs)
cd /opt/progs/COP_LR
**6)** run simulation (it will produce png files for each coupling time step)
./run.sh

A new section (Sec. 6) is also included to the manuscript to discuss interoperability, portability and reproducibility.

**Supplementary videos**

**66)** Useful and nice! Please make sure the video dois are referenced in the text, and add to the videos a link to your publication.

The video DOIs are included into the manuscript and the reference sections.

[revised manuscript text omitted]

---

## Author Comment (AC2) · 2 Nov 2018

**Response to Reviewer II This document is color-coded as follows:**

- Comments by reviewers are in **blue**.
- Our responses are in **black**.
- Blocks of text that is added to the are in **red**
- **The track-changes can be also seen as a supplement document**

**1)** "… Since a key focus on the paper is interoperability afforded by using a standard coupling framework, some additional discussion on details of the software engineering and approach to interoperability could be discussed including more details on the actual interfaces used between components as well as issues related to portability. Follow on work could look at applying the same co-processing component to a completely different model to understand how generic the approach is. A related question is how hard it would be to change out the visualization package itself, since there are a number of packages that offer custom analyses …"

The modeling system is tested in different variety of computing environment ranges from single server with dedicated NVIDIA GPU (M60) to cluster designed to GPU intensive applications (each node has four NVIDIA P100). Using experience to teste the modeling system with a variety different configuration and computing environment, a completely new section (Sec. 6) named as "Discussion of the concepts associated with interoperability, portability, and reproducibility" is added to the manuscript just before result and conclusion section.

Currently, as a part of ongoing national project, we are working on applying model to different use cases that air-sea interaction plays important role such as wind introduced upwelling in Aegean Sea and the relationship between SST anomaly and precipitation in Black Sea. In both cases, different coupled model configurations are used. In the first case, the coupled modeling system configured to use WRF and ROMS but in the second use case WRF and MITgcm configuration is used. The initial results show that the co-processing component is able to render information coming from different model setups. The only requirement is that the NUOPC cap of each different ocean and atmosphere model need to be modified slightly to provide also three-dimensional fields along with two dimensional ones. This also requires to define 3d grid representation in the NUOPC cap of the model component as well as states to allow transferring 3d fields to co-processing component. In theory, using different visualization tool such as VisIt (developed by LLNL, https://visit.llnl.gov) only requires modification in the adaptor code resides between NUOPC cap and visualization tool.

**2)** "… In addition to timing profiles and since the initial motivation was around the problem of data volumes, the paper could benefit from plots describing the amount of data exchanged and used in the in-situ case versus the amount of data that would be required for offline visualization at the same temporal frequency. This would allow an "apples-to-apples" comparison of the online vs. offline approaches.…"

The amount of data exchanged with co-processing component is given in Table 1 for three different visualization pipelines (P1, P2 and P3). In in-situ visualization mode, the data is read from the memory and passed to the ParaView, Catalyst for rendering. In this case, live visualization allows to connect the running simulations and make real-time data analysis of simulation results. If the modeling system configured to run in co-processing mode then rendered image files or data can be stored in the disk. Besides, processing data in co-processing component concurrently with the simulation, the offline visualization (post-processing) uses stored data to make analysis. In in-situ visualization mode, user needs to store variables only for single time snapshot in the memory. Unlike in-situ visualization, offline visualization performed after simulation ends and requires to store data produced by the whole simulation. For example, in case of using offline visualization, 3-days long simulation with 6-minutes coupling interval produces around 160 GB data (720 time step) just for single variable from high-resolution atmosphere component (P1 visualization pipeline). The same analysis can be done with in-situ visualization just by storing single time step, which is the size of 224 MB. Moreover, the netCDF formatted output of RegCM atmosphere model (used in this study) contains 7 x 3d fields and 28 x 2d fields in default configuration and only three-day high-resolution simulation produce around 1.5 TB data in case of using 6-minutes interval. In this case, the data size that needs to be stored in the disk is also depend on used number of active model components, and selected time interval. It is oblivious that the

detailed examination of the simulation results with multiple visualization pipeline requires more variables to be stored in the disk (in VTK binary format) when post-processing method is used. The size of the data also depends on the horizontal and vertical resolution of the model components as well as specified time interval to interact with co-processing component. The desired fps (frame-per-second) in the animation that is produced using output of the co-processing component also defined by the interaction interval (or coupling time step) and usually 20-24 fps is required to have a real world like animation. The following section is added to Section 4.3.

[revised manuscript text omitted]

**Figure 13.** Rendering of three-dimensional vorticity streamlines ($\frac{1}{s}s^{-1}$), total precipitation ($\frac{mm}{day}mmday^{-1}$) and sea surface temperature anomaly (degC) of COP_LR simulation for 28-Aug-2005 00:00 UTC **(a-c)** and 29-Aug-2005 00:00 UTC **(d-f)**. Streamlines are calculated only from the eye of the hurricane. In this case, red and yellow colored forward streamlines  represent cloud liquid water content ($\frac{kg}{kg}kgkg^{-1}$), and blue colored backward streamlines  indicate wind speed ($\frac{m}{s}ms^{-1}$). The  solid yellow line represents the best track of Hurricane Katrina, which is extracted from HURDAT2 database.

[Figure]

**Figure 14.** Same with Fig. 13 but for COP_HR simulation. The comparison of low and  high-resolution model results is shown in the supplemental video.

---

## Author Comment (AC3) · 4 Dec 2018

**Response to Topical Editor This document is color-coded as follows:**

- Comments by topical editor are in **blue**.
- Our responses are in **black**.
- Blocks of text that is added to the are in **red**
- **The track-changes can be also seen as a supplement document**

**Title, Abstract**

**1)** Your argument about keeping "Toward" in the title is fine. However, you don't answer the reviewer's remark about the use of the word "integrated". I agree with the reviewer that use of "integrated" is not appropriate, as it seems contrary to the modularity of the approach. So, what about "Toward modular in-situ visualization in Earth System Models: the regional modeling system RegESM 1.1".

The title is changed as suggested.

**2)** I still propose many language modifications in the attached ManuscriptAnnotated.pdf

All of them is fixed in the latest version of the manuscript.

**43)** To answer the referee's remark, you now mention the "implicit" coupling on l.30 but before defining it; the definition "The main difference between the implicit and semi-implicit coupling type is that the models interact on different time scales in implicit coupling scenarios. » comes only after but is not clear at all. Contrary to what was asked by the referee, I suggest to remove the text about the implicit coupling as it is not an option in RegESM (if I understand well).

The text related with implicit coupling is removed.

**46)** It is OK for me not to describe how ESMF handles this. But your description on p.9, l.19-32 is very hard to follow. In particular, I don't understand at all what "According to the algorithm, the mapped grid points have same land-sea mask type in both model components (i.e., both are sea or land). On the other hand, the land-sea mask type does not match completely in the case of unmapped grid points" means. Can you try to review and simplify it, maybe giving the essence of the method and not all the technical steps? Also on Fig. 4, a "on" is missing between "only" and "grid" in "All interpolations are performed only grid points over SEA".

The text is simplified (please see track changes) and Figure 4 is also fixed.

**47)** I am sorry to say that I think that the added paragraph does not help understanding the algorithm.

Please refer to the previous item. The unnecessary implementation details are removed from the paragraph. The algorithm is mainly very simple and perform two different interpolation (bilinear and nearest-neighbor) to the exchange field and compare the results to find mapped (land-sea mask is same in both source and destination grid) and unmapped (mismatch in land-sea mask) grid points. Then, using this information, it performs interpolation and extrapolation to transfer exchange field to the destination grid.

**49)** ... A visualization pipeline integrates a data flow network in which computation is described as a collection of executable modules that are connected in a directed graph representing how data moves between modules (Moreland, 2013). There are three types of modules: sources (file readers and synthetic data generators), filters (transforms data), and sinks (file writers and rendering module that provide images to a user interface) in the visualization pipeline. ...

There is some incoherency in the first paragraph of section 3. The paragraph starts by describing the "conventional co-processing" and without transition discusses the NUOPC cap (which, if I understand well, is part of the novel approach and not the conventional one); please clarify.

The paragraph is modified slightly and the wrongly placed "NUOPC cap" is replaced with the "simulation code". The first paragraph in Section 3 describes the conventional co-processing systems and second one is for ESMF integrated one.

Also, in the abstract and introduction, you use the word "conventional" associated to "post-processing", which may lead to some confusion. I would advice to use "traditional" instead of "conventional" when qualifying the post-processing and keep "conventional" when qualifying the co-processing.

The text is modified based on your suggestion. Please refer to the track changes.

**52)** Can you specify how you calculate the overhead? Do you compare the CPL wall clock-time to the sum of the standalone OCN and ATM wall clock time as they run sequentially? Please clarify.

Yes. It is correct. The text is modified as "… The overhead is calculated by comparing the CPL wall clock-time to the sum of the standalone OCN and ATM wall clock time as they run sequentially. …"

**53)** Please add the justification of the vertical interpolation in the manuscript.

The following sentence is added to the end of the Section 3. "… In this design, the vertical interpolation is introduced to have a consistency in the vertical scales and units of the data coming from the atmosphere and ocean components. …"

**54)** I agree with the reviewer that 40% is very significant. I suppose it is linked to the fact that the components are all run sequentially (if I understand well). You should say something about the overhead if the co-processing was run concurrently; more processes would be needed but the wall-time would probably not increase.

Yes. The components are run in an order using same compute resource (or cores) and components waits for co-processing components to render data. In a computing environment without GPU support, the rendering of the information in co-processing component takes more time and rest of the model components waits for their order and co-processing component become a bottleneck for whole modeling system. It is clear that having co-processing component that runs concurrently and process the data along with the simulation will boost the performance of the modeling system. The following text is added to the manuscript "In this case, the components are all run sequentially, and the performance of the co-processing component becomes a bottleneck for the rest of the modeling system especially for the computing environment without GPU support like the system used in the benchmark simulations. It is evident that if the co-processing were run concurrently in a dedicated computing resource, the overall performance of the modeling system would be improved because of the simultaneous execution of the physical models and co-processing components."

**59)** I cannot really comment the figures, as I don't understand the algorithm, see my remark above.

The paragraph is simplified and low-level design details are removed.

**61)** You should mention somewhere in the text that figures 9 and 10 show the wall clock time and the speed-up. Please define precisely how you calculate the speed-up either in the text or in the captions.

The following text in Section 4.3 "… The benchmark results of standalone model components (ATM and OCN) can be seen in Fig. 9 …" is changed as "… The measured wall clock time and the calculated speed-up of standalone model components (ATM and OCN) can be seen in Fig. 9 …". I also included "Similar to the benchmark results of the standalone model components, the measured wall clock time and the calculated speed-up of the coupled model simulations are also shown in Fig.~\ref{fig:10}." to the text. Additionally, calculation of the speed-up is included as caption.

**Additional remarks**

- For OASIS3-MCT, please cite: "Craig A., Valcke S., Coquart L., 2017: Development and performance of a new version of the OASIS coupler, OASIS3-MCT_3.0, Geoscientific Model Development, 10, pp. 3297-3308, doi:10.5194/gmd-10- 3297-2017"

  It is added to the references and also the text.

- Many places in the text, you use « ParaView, Catalyst » to design the ParaView co- processing plugin. Please use « ParaView Catalyst » without the comma or « ParaView/Catalyst »

  They are fixed and replaced by ParaView/Catalyst.

- COP is used sometimes to design the co-processing component and sometimes the three-component coupled system simulations. This is confusing. Please use "co-processing component" for the component and keep COP to design only the three-component coupled system simulations. Please define COP and CPL the first time it appears in the text.

  The manuscript is modified based on your suggestion. For component "co-processing component" and for three-component simulations "COP" are used. In the Section 4.3, the following text "… in the coupled model simulations (CPL and COP)" is changed as "… in the coupled model simulations; CPL (two component case: atmosphere-ocean) and COP (three-component case: atmosphere, ocean and co-processing component) …".

- Fig. 7 is misleading. It looks like the ATM and OCN components are coupled through the co-processing component. If I understand well, this figure should just illustrate the interaction between ATM and the co-processing component on one side, and the interaction between OCN and the co-processing component on the other side. If I am right could it be possible to split the figure into two parts so to avoid the confusion?

  The figure aims to demonstrate grid transfer feature and mapping domain decomposition configuration for each model components in co-processing side. In my opinion, splitting figure does not help because there is no direct interaction among model components and all information need to be passed through the driver component. This will still cause confusion. So, i prefer to keep only one figure but in this case, i modified the figure to include also driver component to be clear (please see new version of Fig. 7).

- In section 4.2, an HR (inner) atmosphere is nested in an LR (outer) atmosphere. In section 4.3, HR and LR are used, if I understand well, for two different atmospheres covering the whole atmospheric domain (no nesting); this is confusing. Can you keep HR and LR for section 4.3 only, and change the wording when referring to the HR-inner atmosphere nested in the LR-outer atmosphere in section 4.2?

  In the use case and benchmark analysis, we used offline nesting approach (i changed word "one-way" to "offline" in Sec. 4.2) for atmosphere component. The HR domain is nested inside of LR domain and retrieve initial and boundary condition form LR model. This is already explained in Fig. 4.2. The Section 4.2 and 4.3 are checked again to prevent any confusion.

- P.14 , l.4: "their THREDDS server"; can you explain what THREDDS means or give a link?

  Following sentence is added "… their THREDDS (Thematic Real-time Environmental Distributed Data Services) data server (TDS). THREDDS is a service that aims to provide access to an extensive collection of real-time and archived datasets, and TDS is a web server that provides metadata and data access for scientific datasets, using a variety of remote data access protocols. The ocean model …"

- P.14, l.30: You refer to section 2.5 for details on the limitation of the co-processing about its sequential type execution, but I don't see where this is detailed in section 2.5. Can you clarify or point me to the exact paragraph?

Section 2.5 includes detailed information about sequential and concurrent type execution of model components not the limitation of the co-processing component under sequential type execution. To prevent any confusion, the "… (see Section 2.5) for more information) …" is removed from the manuscript.

- Table 1: Pipeline details are not readable. Table 1 caption: please remove "in" in "… are shown in here …"

It is hard to use bigger figures for the visualization pipeline shown as a screenshot of ParaView pipeline browser due to the size of the table and used number of columns to show three pipeline in the same time. As a workaround, the visualization pipelines will be included as supplementary material in the final version of the manuscript (see Fig. 1-3 in the supplementary material). The caption is modified as suggested and "… The visualization pipelines are also given as supplementary material …"

- P.15, l.6-8: I don't understand why you write "It is also shown that around 588 processors, which is the highest available compute resource, the communication among the processors dominate the benchmark results and even HR case does not gain further performance": the HR curve (triangles) does not flatten as the LR curve. Please clarify.

The sentence "… results and even HR case does not gain further performance …" is changed as "… results of LR case, but it is not evident in HR case and scales very well without any performance problem …"

- Fig. 9: I don't understand what the envelope represents and what do you mean by "as a line". Why isn't the best configuration the lower limit of the envelope?

In Fig. 9b, the envelope represents the timing results that are done using same number of core but different two-dimensional decomposition configuration. For example, the possible two-dimensional decomposition configurations can be 1x28, 2x14, 4x7, 7x4, 14x2 and 28x1 for 28 cores. The figure is modified and now the solid line shows the lower limit of the envelope in wall-clock time measurements and upper limit in scaling results. The numbers represent the best tile configuration in timing results. The caption of the Fig. 9 is also modified as "Benchmark results of standalone (a) atmosphere (ATM; both LR and HR) and (b) ocean (OCN) models. Note that timing results of the atmosphere model are in log axes to show both LR and HR cases in the same figure. The black lines represent measured wall clock times in second and red lines show speed-up. The envelope represents the timing and speed-up results that are done using the same number of cores but different two-dimensional decomposition configuration. The best two-dimensional decomposition parameters are also shown in the timing results for the ocean model case."

- P. 15, l.23: I don't understand what "acceptable when increased number of MPI communication between the components are considered" means in this context. Is it a justification/explanation of the 5-10% overhead? Please rephrase.

Yes. It is just justification for slower model performance (5-10%) along with increased number of cores. The overhead also includes overhead of the driver component (data transfer, synchronization and remapping). The sentence is also modified as "… The extra overhead is mainly due to the interpolation (sparse matrix multiply performed by ESMF) and extrapolation along the coastlines to match land-sea masks of the atmosphere and ocean models and fill the unmapped grid points to exchange data (Fig. 4) and slightly increases along with increased number of cores as well as number of MPI communication between the model components (Fig. 9 and 10a). …"

- I think you don't need to put a capital letter to each word in titles in English.

  They are fixed as suggested.

- Color scales in Fig 12, 13 and 14 are not readable.

  Again, this is mainly related with the limited space. It is possible to zoom the figures to see the color scales for the online version. Also, they are included into the supplementary material as an individual figures to make them more readable and larger. The following text is also added to the captions of Fig. 13 and 14 "… The larger versions of figures are also given as supplementary material …"

- P.19, l.15: Please rephrase "have higher water content in a decreasing trend with height and spatial distribution" as it does not seem grammatically correct to me.

[revised manuscript text omitted]

---

## Editor Decision (ED1)

Dear Author,

Thank you for submitting your updated manuscript and for carefully replying to the reviewers' remarks. I consider that you have answered most of them but I still have some questions regarding some remarks of the first reviewer.

I also have some additional remarks that you will find here below.

Finally, while reading the manuscript, I could not resist making many propositions to improve the style of the text that you will also find starting on p.7 below. To construct this, I copied and pasted the content of your pdf manuscript in a Word file and made my changes in revision mode, so that you can easily identify them. The lines of the original manuscript appear separately, sorry for this but I think this was the most efficient way to propose many minor modifications. I hope you will find this useful to improve the style of the manuscript.

In the following, the reviewers remarks are in blue, your replies in black and my additional questions or comments in orange.

**1) Reviewer 1**

**Title, Abstract**

1) title Prepositions/articles missing. Also, I don't agree with "integrated" but suggest to use the word "modular". As the visualization has been proven, it is not a perspective "towards", but already production ready, so don't use "towards". I would rephrase the title (this is a mere suggestion) as "Co-processing online visualization: the Regional Earth System RegESM version 1.1".

In the current implementation, the adaptor code between NUOPC cap and ParaView, Catalyst is just a prototype that supports only for regional modelling cases with limited spatial grid support (structured and curvilinear grids). It still needs additional work to be more generic and adaptor code needs to be extended to include support for different mesh types such as unstructured grid for both global and regional applications. These are not implemented yet and needs to design adaptor code carefully for more generic applications. This will be done in the future.

Your argument about keeping "Toward" in the title is fine. However, you don't answer the reviewer's remark about the use of the word "integrated". I agree with the reviewer that use of "integrated" is not appropriate, as it seems contrary to the modularity of the approach. So what about "Toward modular in-situ visualization in Earth System Models: the regional modeling system RegESM 1.1".

2) abstract Fix language issues, such as: articles, avoid "being" before participles, check use of due to/because of, avoid overly use of "used", avoid "have to", do processes literally "move"? Avoid use of "basically", and many more. It is a major revision task to carefully check each sentence

Abstract section is modified based on your suggestions.

I still propose many language modifications in the attached ManuscriptAnnotated.pdf .

**43)** p8 l18f You name stability as an advantage of semi-implicit coupling. Also add draw-backs of semi- implicit and advantages of implicit time stepping.

The paragraph is extended to include more information about different coupling schemes.

To answer the referee's remark, you now mention the "implicit" coupling on l.30 but before defining it; the definition "The main difference between the implicit and semi-implicit coupling type is that the models interact on different time scales in implicit coupling scenarios. » comes only after but is not clear at all. Contrary to what was asked by the referee, I suggest to remove the text about the implicit coupling as it is not an option in RegESM (if I understand well).

**46)** p8 l31f The description of regridding could be a subsection, section 2.6 "Connector". How do other coupled model systems handle this (any references available?). Recently, I implemented into MOSSCO a very similar scheme using bilinear for interpolation and nearest neighbour for extrapolation, so I believe this is a very good way to handle this (but it is not published yet). Possibly, describe how ESMF handles this rather than pointing vaguely to ongoing development.

I prefer to keep it as it is. The special interpolation implemented in RegESM modeling system is a part of driver.

It is OK for me not to describe how ESMF handles this. But your description on p.9, l.19-32 is very hard to follow.

In particular, I don't understand at all what "According to the algorithm, the mapped grid points have same land-sea mask type in both model components (i.e., both are sea or land). On the other hand, the land-sea mask type does not match completely in the case of unmapped grid points" means.

Can you try to review and simplify it, maybe giving the essence of the method and not all the technical steps? Also on Fig. 4, a "on" is missing between "only" and "grid" in "All interpolations are performed only grid points over SEA".

**47)** p9 l14 Align your description with the terms used in Figure 4 (see comments below on this figure). We used field names (Field_A, Field_B and Field_C) in also manuscript. The paragraph is also extended to describe the algorithm clearly.

I am sorry to say that I think that the added paragraph does not help understanding the algorithm.

**49)** p9 l28 Define "conventional" or give a reference to such approaches. Define pipeline in this context.

The conventional co-processing enabled simulation systems interacts with single physical model component such as atmosphere along with co-processing support. This is defined also in text and references are added.

... A visualization pipeline integrates a data flow network in which computation is described as a collection of executable modules that are connected in a directed graph representing how data moves between modules (Moreland, 2013). There are three types of modules: sources (file readers and synthetic data generators), filters (transforms data), and sinks (file writers and rendering module that provide images to a user interface) in the visualization pipeline. ...

There is some incoherency in the first paragraph of section 3. The paragraph starts by

describing the "conventional co-processing" and without transition discusses the NUOPC cap (which, if I understand well, is part of the novel approach and not the conventional one); please clarify.

Also, in the abstract and introduction, you use the word "conventional" associated to "post-processing", which may lead to some confusion. I would advice to use "traditional" instead of "conventional" when qualifying the post-processing and keep "conventional" when qualifying the co-processing.

**52)** p14 l26 The "driver component introduces additional 5-10%". In ESMF documentations, usually below 5% overhead is assumed. this makes sense as you to the SMM twice for each interpolation of mask extrapolation. It would be interesting to see whether the generic implementation of extrapolation is more efficient in ESMF 8 (out of scope of this MS).

I agree. As far as i know, the ESMF developers was trying to find a way to combine weight matrix of multiple interpolation (including extrapolation) into a single one. Similar methodology is using in transformation (scale, rotation, translation etc.) matrixes using homogeneous coordinates but i am not sure it could be used in regridding or not.

Can you specify how you calculate the overhead? Do you compare the CPL wall clock-time to the sum of the standalone OCN and ATM wall clock time as they run sequentially? Please clarify.

**53)** p14 l35 " includes vertical interpolation to map data from sigma coordinates to height coordinates". Why is this interpolation step necessary? In principle, the visualization should equally work on sigma layers. And is the interpolation performed within the ESMF connector, or within a Paraview pipeline?

I did not try yet but yes the visualization works with sigma coordinates without any problem. The vertical interpolation is introduced to have a consistency in the data coming from atmosphere and ocean. By this way, the vertical scales of the data can be compared without any problem. In general, both atmosphere and ocean components use sigma coordinates with different definitions. The interpolation is done in ESMF cap not in ParaView side. It could be done in ParaView side also but in this case all grid related parameters such as stretching functions etc. need to be passed to ParaView, Catalyst and special ProgrammableFilter need to be developed.

Please add the justification of the vertical interpolation in the manuscript.

**54)** p15 l2 the computational demand for visualization "require 10-40% extra" is rather high and its evaluation would probably change depending on whether it is more 10% (acceptable) or 40% (not acceptable). The benchmark using software rendering with Mesa is not ideal to demonstrate this, especially as many new HPC systems come with dedicated GPU nodes that could much reduce the visualization overhead.

It is clear that Mesa is not right way to measure the performance of the co-processing component. The overall performance of the used computing system dominates the results in our case but we don't have accesses to a system with GPU accelerators. In the early stage of the performance benchmark, Dell provided us a grant in newly installed University Manchester system (10000 core/hour) but the system is mainly designed for GPU intensive and AI type researches and the nodes have small number of core (10 cores) but four NVIDIA P100 GPU. In fact, our test simulations are CPU intensive and

requires high amount of compute resource (or cores) for the high-resolution simulations and rendering only done on a single GPU (limitation of ParaView, Catalyst and used filters). As a result, couple of benchmark run spend all allocated resources (unused GPU resources was also accounted) and the system was not used for the benchmark. There was also issue related with buggy NVIDIA driver and ParaView protobuf library. In the future, along with the development of m-VTK (GPU accelerated filters) it could be possible to use multi-GPU systems.

I agree with the reviewer that 40% is very significant. I suppose it is linked to the fact that the components are all run sequentially (if I understand well). You should say something about the overhead if the co-processing was run concurrently; more processes would be needed but the wall-time would probably not increase.

**59)** Figure 4 Rather unreadable. Please redesign. (1) Avoid bridges (this is possible by routing the arrow "Create ESMF FIELD_B" north to "Create ESMF FIELD_C". (2) don't use same line thickness/color for outlining the boxes around src/dst grid, rather make them visually less intrusive to highlight the flowchart. (3) Abstract away 1.0, LARGEST, "over sea" or other value-based masking, if possible.

The figure is modified. The hop is solved by changing the position of the arrow. The dashed line is used in the boxes and color is changed to grey to move focus to workflow. It is hard to abstract used values and mask values. The algorithm is tested with this configuration and it is better to keep them as it is to have a reproducible result (at least for finding mapped and unmapped grid points). I am also using same methodology using NCL ESMF interface to create forcing for ocean models and it works better than standard extrapolation method (based on solution of Possion Eq.) provided by NCL itself.

I cannot really comment the figures, as I don't understand the algorithm, see my remark above.

**61)** Figures 9/10 It is very difficult to compare ATM versus OCN performance using different y scales. I recommend to use the same ylog-scale for all panels in figs 9 and 10. Also try loglog. Tile sizes are hard to read, please move text from lines. Calculate all speedup relative to 140 cores, so that you don't have different reference points. Explain model versions like RegCM_r6274, delete UHeM. Avoid overlapping graphics and text.

The speed-up calculation of coupled model simulations starts from 140 core because this is the minimum number of cores that allows to run the modeling system. For standalone simulations, it is possible to run the model with 28 cores (single node). As a result, it was used 28 cores for standalone and 140 cores for coupled model simulations to calculate the speedup. The plots are modified to have consistent x and y axis (log scale) as well as number of cores used in the speed-up calculation (140 cores) except plot for standalone ocean model. In this case, using same log scale for it does not give good results as expected for example the effect of tile configuration cannot be seen clearly because the benchmark results fall into a very narrow interval. To that end, i prefer to keep results of standalone ocean simulations as it is. I also increased font sizes and move the text little bit far from the markers for standalone ocean model plot. The texts that indicate model version and name of the computing system that is used in the benchmark are removed because they are already mentioned in the manuscript.

You should mention somewhere in the text that figures 9 and 10 show the wall clock time and the speed-up. Please define precisely how you calculate the speed-up either in the text or in the captions.

**Additional remarks**

- For OASIS3-MCT, please cite: "Craig A., Valcke S., Coquart L., 2017: Development and performance of a new version of the OASIS coupler, OASIS3-MCT_3.0, Geoscientific Model Development, 10, pp. 3297-3308, doi:10.5194/gmd-10-3297-2017"

- Many places in the text, you use « ParaView, Catalyst » to design the ParaView co-processing plugin. Please use « ParaView Catalyst » without the comma or « ParaView/Catalyst »

- COP is used sometimes to design the co-processing component and sometimes the three-component coupled system simulations. This is confusing. Please use "co-processing component" for the component and keep COP to design only the three-component coupled system simulations. Please define COP and CPL the first time it appears in the text.

- Fig. 7 is misleading. It looks like the ATM and OCN components are coupled **through** the co-processing component. If I understand well, this figure should just illustrate the interaction between ATM and the co-processing component on one side, and the interaction between OCN and the co-processing component on the other side. If I am right could it be possible to split the figure into two parts so to avoid the confusion?

- In section 4.2, an HR (inner) atmosphere is nested in an LR (outer) atmosphere. In section 4.3, HR and LR are used, if I understand well, for two different atmospheres covering the whole atmospheric domain (no nesting); this is confusing. Can you keep HR and LR for section 4.3 only, and change the wording when referring to the HR-inner atmosphere nested in the LR-outer atmosphere in section 4.2?

- P.14 , l.4: "their THREDDS server"; can you explain what THREDDS means or give a link?

- P.14, l.30: You refer to section 2.5 for details on the limitation of the co-processing about its sequential type execution, but I don't see where this is detailed in section 2.5. Can you clarify or point me to the exact paragraph?

- Table 1: Pipeline details are not readable. Table 1 caption: please remove "in" in "... are shown in here ..."

- P.15, l.6-8: I don't understand why you write "It is also shown that around 588 processors, which is the highest available compute resource, the communication among the processors dominate the benchmark results and even HR case does not gain further performance": the HR curve (triangles) does not flatten as the LR curve. Please clarify.

- Fig. 9: I don't understand what the envelope represents and what do you mean by "as a line". Why isn't the best configuration the lower limit of the envelope?

- P. 15, l.23: I don't understand what "acceptable when increased number of MPI communication between the components are considered" means in this context. Is it a justification/explanation of the 5-10% overhead? Please rephrase.

- I think you don't need to put a capital letter to each word in titles in English.

- Color scales in Fig 12, 13 and 14 are not readable.

- P.19, l.15: Please rephrase "have higher water content in a decreasing trend with height and spatial distribution" as it does not seem grammatically correct to me.

[revised manuscript text omitted]

Sophie Valcke 23/11/y 16:57

Sophie Valcke 23/11/y 16:58

Sophie Valcke 23/11/y 16:59

Sophie Valcke 23/11/y 16:59

Sophie Valcke 23/11/y 16:59

Sophie Valcke 23/11/y 17:00

Sophie Valcke 23/11/y 17:00

Sophie Valcke 23/11/y 17:00

Sophie Valcke 23/11/y 17:00

Sophie Valcke 23/11/y 17:00

Sophie Valcke 23/11/y 17:00

Sophie Valcke 23/11/y 17:01

Sophie Valcke 23/11/y 17:01

Sophie Valcke 23/11/y 17:01

Sophie Valcke 23/11/y 17:01

Sophie Valcke 23/11/y 17:02

Sophie Valcke 23/11/y 17:02

Sophie Valcke 23/11/y 17:02

Sophie Valcke 23/11/y 17:03

Sophie Valcke 23/11/y 17:03

Sophie Valcke 23/11/y 17:04

Sophie Valcke 23/11/y 17:04

Sophie Valcke 23/11/y 17:04

Sophie Valcke 23/11/y 17:04

Sophie Valcke 23/11/y 17:05

[revised manuscript text omitted]

Sophie Valcke 23/11/y 17:21

Sophie Valcke 23/11/y 17:23

Sophie Valcke 23/11/y 17:23

Sophie Valcke 23/11/y 17:23

Sophie Valcke 23/11/y 17:24

Sophie Valcke 23/11/y 17:24

Sophie Valcke 23/11/y 17:24

Sophie Valcke 23/11/y 17:24

Sophie Valcke 23/11/y 17:24

Sophie Valcke 23/11/y 17:25

Sophie Valcke 23/11/y 17:25